# Mathesis: Towards Formal Theorem Proving from Natural Languages

**Xuejun Yu**[1,*] **Jianyuan Zhong**[2,3,*] **Zijin Feng**[2,*] **Pengyi Zhai**[1] **Roozbeh Mohit**[2]

**Wei Chong Ng**[1], **Haoxiong Liu**[2], **Ziyi Shou**[1] **Jing Xiong**[2] **Yudong Zhou**[1] **Claudia Beth Ong**[1]

**Austen Jeremy Sugiarto**[1] **Yaoxi Zhang**[1] **Wai Ming Tai**[1] **Huan Cao**[1] **Dongcai Lu**[1]

**Jiacheng Sun**[2] **Qiang Xu**[3] **Shen Xin**[1,†] **Zhenguo Li**[2,†]

[1]Huawei Celia Team
[2]Huawei Foundation Model Department
[3]The Chinese University of Hong Kong
{Li.Zhenguo,shenxin19}@huawei.com

## Abstract

Recent advances in large language models (LLMs) show strong promise for formal reasoning. However, most LLM-based theorem provers remain constrained by the need for expert-written formal statements as inputs, limiting their applicability to real-world problems expressed in natural language. We address this gap by focusing on autoformalization, the task of translating informal problems into formal statements. We propose Mathesis, the first pipeline for the systematic study of formal theorem proving from natural language. It contributes the first autoformalizer trained with reinforcement learning, which integrates syntactic, semantic, and prover feedback as reward signals to yield accurate and verifiable formalizations. This is further supported by our novel LeanScorer framework for evaluating semantic correctness. To assess real-world applicability, we introduce Gaokao-Formal, a benchmark of 495 complex proof problems from the college entrance exams. Experiments demonstrate that our autoformalizer improves pass rates by 45% on Gaokao-Formal and 6% on MiniF2F compared to state-of-the-art baselines. Paired with provers, our autoformalizer consistently enhances proving accuracy, including a 42% gain for DeepSeek-Prover-V2 on Gaokao-Formal. Our code is available at https://github.com/Huawei-AI4Math/Mathesis.

## 1 Introduction

The emergence of reasoning abilities in large language models (LLMs) has opened new frontiers in automated mathematics (Yang et al., 2025a). Recent automatic theorem provers (ATPs) leverage formal verification systems, such as Lean (mathlib Community, 2020; Moura & Ullrich, 2021), Isabelle (Paulson, 1994), and Coq (Huet et al., 1997), to enable formal reasoning. Formal reasoning starts with a clear formal problem statement, followed by the generation of mechanically verifiable proofs in formal languages. This approach ensures greater reliability and verifiability. Notable models in this field, including Deepseek-Prover-V2 (Ren et al., 2025), Kimina-Prover (Wang et al., 2025), and Goedel-Prover (Lin et al., 2025), are currently state-of-the-art on benchmarks such as MiniF2F (Zheng et al., 2021), proofNet (Azerbayev et al., 2023), and PutnamBench (Tsoukalas et al., 2024), where the input problem statements have been formalized by human experts.

However, real-world mathematical problems are typically written in natural language, which prohibits the direct use by ATPs. Traditionally, manual formalization ensures faithful translation of problems but requires significant effort and expertise before ATPs can solve them. In this paper, we study automatic **formal theorem proving from natural language**. The task begins with a given natural language (NL) problem statement and translates it into formal language (FL), followed by the generation of a formal proof. A critical step in this task is autoformalization–the process of automatically translating informal mathematics into formal language (Gao et al., 2024)–which can

---

* Equal Contribution. † Corresponding authors.

**Informal Statement:** Let $S_n$ denote the sum of the first $n$ terms of the sequence $a_n$. Given that $\frac{2S_n}{n} + n = 2a_n + 1$, prove that $\{a_n\}$ is an arithmetic sequence.

**Two Formal Statement Cases:**
```
theorem case_one (a :  ℕ → ℝ)
 (ha :  ∃ d, ∀ n, a (n + 1) = a n + d)  ⇒  Erroneously includes the desired goal in

 assumptions
(h :  ∀ n, 2 * (∑ i in Finset.range n, a i) / n + n = 2 * a n + 1) :
∃ d, ∀ n, a (n + 1) = a n + d := by sorry

theorem case_two (a :  ℕ → ℝ) (S : ℕ → ℝ)
 (hS : ∀ (n :  ℕ), n ≥ 1, S n =  ∑ k in Finset.range n, a k)  ⇒

 ∑ k in Finset.Icc 1 n, a k
(h :  ∀ (n :  ℕ), n ≥ 1, 2 * S n / (n :  ℝ) + (n :  ℝ) = 2 * a n + 1) :
∃ (d :  ℝ), ∀ (n :  ℕ), n ≥ 1, a (n + 1) = a n + d := by sorry
```

Figure 1: Illustrative examples of incorrect formalization. Case 1 mistakenly includes the goal as an assumption, resulting in a circular yet technically provable formalization that is mathematically invalid. Case 2 mistranslates the summation range, leading to an incorrect formal statement that is both unprovable and misaligned with the informal input.

significantly impact the success of proving due to errors introduced during formalization. Figure 1 illustrates two common examples, highlighting how improper formalization can yield misleading proof successes or render problems unprovable.

Despite its importance, the task remains understudied, particularly in terms of semantic evaluation and the availability of a powerful autoformalizer. On the evaluation side, existing methods rely on Lean syntactic compilation checks and basic binary LLM judgments (Lin et al., 2025; Gao et al., 2024; Ying et al., 2024), which fail to capture nuanced semantic errors, resulting in the absence of fine-grained evaluation and limiting the performance of formal theorem proving from natural language. Moreover, most existing formal benchmarks are not designed to assess the quality of autoformalization. These benchmarks might allow easy passage of basic checks but fail to reveal semantic errors, necessitating harder-to-formalize benchmarks for rigorous evaluation. Specifically, some benchmarks simplify the original problems in ways that alter their intent, or exclude problem types that are challenging to formalize, such as those involving geometry, combinatorics (Zheng et al., 2021), and word problems (Azerbayev et al., 2023). On the autoformalizer side, recent methods (Jiang et al., 2023; Gao et al., 2024; Liu et al., 2025b; Lin et al., 2025) fine-tune LLMs on paired informal and formal statements (a.k.a parallel statements (Jiang et al., 2023; Liu et al., 2025b)) for higher quality, with Kimina-Autoformalizer (Wang et al., 2025) achieving state-of-the-art performance via expert iteration. However, these training approaches lack dynamic learning from direct feedback on both syntactic and semantic correctness.

In this paper, we present Mathesis (**M**ulti-domain **A**utoformalization **T**hrough **He**uristic-guided **S**yntactic and **S**emantic Learning), an autoformalization-driven formal theorem proving pipeline that solving natural language problems. To our knowledge, we are the first to systematically study the entire workflow from natural language input to formal proof generation—a critical yet previously overlooked aspect in the community. At the core of Mathesis is Mathesis-Autoformalizer, the first autoformalization framework trained via online reinforcement learning (RL) enhanced by a novel Hierarchical Preference Optimization (HPO) mechanism. By incorporating Lean compilation and semantic verification into the RL reward function while learning prover preferences through HPO, Mathesis significantly enhances formalization quality and achieves state-of-the-art performance. To enable rigorous and nuanced evaluation of formalization quality, we propose LeanScorer, a novel semantic evaluation framework designed to capture subtle errors beyond binary correctness checks. We further introduce Gaokao-Formal, a challenging benchmark of 495 proof problems, spanning a wide range of mathematical domains. Our key contributions are as follows.

- We introduce Mathesis-Autoformalizer, the first autoformalizer trained via online reinforcement learning with rewards signals for syntactic validity, semantic correctness, and prover feedback. It improves pass rates by 22 percentage points (45% relative gain) on Gaokao-Formal and 5 points (6% gain) on MiniF2F over the state-of-the-art baseline.

- We propose LeanScorer, a novel semantic evaluation framework that combines LLM-based analysis with the Sugeno Fuzzy Integral for nuanced assessment of formalizations. LeanScorer achieves a 0.92 F1 score, outperforming prior approaches LLM-as-a-Judge by 7 percentage points and Re-informalization by 27 points on human-annotated data.

- We provide the first systematic study of formal theorem proving from natural language. Extensive experiments show that our autoformalizer consistently improves prover accuracy, with gains up to 122% on Gaokao-Formal and 98% on MiniF2F.

## 2  RELATED WORK

**Formal Reasoning**  Recent advancements have produced powerful LLM-based automated theorem provers for proof assistants like Lean 4, including DeepSeek-Prover-V2 (Ren et al., 2025), Kimina-Prover (Wang et al., 2025), and Goedel-Prover (Lin et al., 2025), alongside many advanced algorithms for proof search (Liang et al., 2025; Xin et al., 2025; Li et al., 2024; Liu et al., 2025a; Yang et al., 2025b). These systems demonstrate strong formal-to-formal (F2F) reasoning capabilities, where both input statements and output proofs are expressed in formal language. Correspondingly, benchmarks such as MiniF2F (Zheng et al., 2021), PutnamBench (Tsoukalas et al., 2024), and FIMO (Liu et al., 2023) are designed to evaluate such F2F reasoning ability using well-formalized problem statements. However, it leaves a critical gap in the study of formal theorem proving from natural language, which requires first formalizing the input informal mathematical statements and then proving them. Our work addresses this gap by introducing a complete pipeline for formal theorem proving from natural language, grounded in autoformalization. While some other works (Zhao et al., 2023; Wang et al., 2023; Jiang et al., 2022; Cabral et al., 2025), such as Lego-Prover (Wang et al., 2023), also target informal-to-formal reasoning, they require an additional informal proof sketch as input. In contrast, our approach performs fully automatic, autoformalization-driven formal theorem proving, starting solely from informal statements to whole-proof generation (Xin et al., 2024), making direct comparisons inappropriate.

**Autoformalization**  Autoformalization, the process of formalizing informal mathematics into formal language , is essential for bridging the NL-FL gap. Prior work includes prompting pre-trained LLMs (Wu et al., 2022; Azerbayev et al., 2023; Poiroux et al., 2024; Patel et al., 2023) and fine-tuning models on static NL-FL pairs (Cunningham et al., 2023; Jiang et al., 2023; Lu et al., 2024b; Gao et al., 2024; Liu et al., 2025b), with systems like Kimina-Autoformalizer (Wang et al., 2025) achieving notable success. However, these approaches often lack dynamic learning from direct feedback on syntactic and semantic correctness, and their evaluation has typically relied on binary compilation checks or basic LLM judgments (Peng et al., 2025; Lin et al., 2025; Liu et al., 2025b), which may not capture nuanced misalignments.

Semantic evaluation itself remains a key bottleneck. Training-free methods such as LLM-as-a-Judge (Lin et al., 2025; Gao et al., 2024) and Re-informalization (Ying et al., 2024) are most widely used, but they offer only indirect signals of semantic correctness and can be inconsistent. In contrast, FormalAlign (Lu et al., 2024a) offers a training-based alternative by fine-tuning an alignment model to compute likelihood- and representation-based scores, though such trained evaluators may generalize less effectively to out-of-distribution mathematical problem types.

To address these limitations, we introduce *Mathesis-Autoformalizer*, which, to our knowledge, is the first autoformalizer to leverage online reinforcement learning and the feedback from the prover for improved accuracy and robustness. Concurrently, our *LeanScorer* framework provides a more fine-grained evaluation of autoformalization quality, moving beyond simple binary pass/fail metrics.

**Reinforcement Learning Fine-Tuning (RLFT)**  Reinforcement learning has proven highly effective for enhancing LLM capabilities in complex reasoning (Anthropic, 2024; DeepSeek-AI & Anonymous Contributors, 2025). The autoformalization task is well-suited for RL, as syntactic validity (from a Lean verifier) and semantic correctness (e.g., assessed by an LLM judge or *LeanScorer*) can serve as direct reward signals. Despite this clear potential, the application of outcome-based RL techniques to specifically optimize these syntactic and semantic properties in autoformalization models has been largely underexplored in the literature. Our *Mathesis-Autoformalizer* pioneers this direction by employing Group Relative Policy Optimization (Shao et al., 2024) with a carefully designed composite reward function (in Section 3.1). This approach

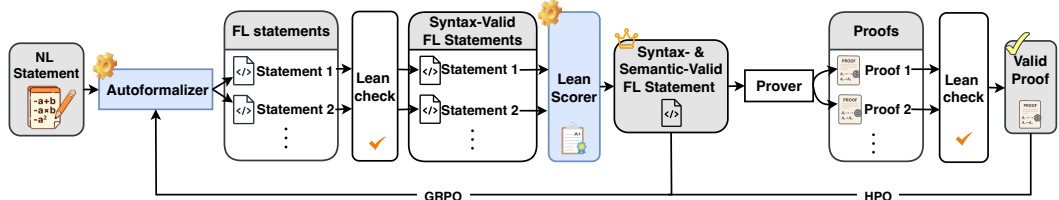

Figure 2: Overview of our autoformalization-driven formal theorem proving pipeline.

allows the model to iteratively refine its ability to generate syntactically correct and semantically faithful formalizations, addressing a key gap in existing autoformalization methodologies.

# 3 MATHESIS: AUTOFORMALIZATION-DRIVEN FORMAL PROVING

The core objective of this paper is to tackle the task of formal theorem proving from natural language (defined in Section 1) by enabling automated formal reasoning directly from informal natural language inputs. To this end, as illustrated in Figure 2, we propose a structured, multi-stage pipeline built upon an autoformalization-driven approach. It consists of three major stages: autoformalization, validation, and proving. The pipeline begins with a given NL problem statement, which is first processed by an autoformalizer to generate candidate formal statements in Lean 4. This stage is powered by our *Mathesis-Autoformalizer*, a model that achieves state-of-the-art performance in formalization (see Section 3.1). These candidates are then evaluated in the validation stage, which includes syntactic verification via the Lean compiler and semantic evaluation. For this purpose, we introduce a novel evaluation framework, LeanScorer, and a new benchmark *Gaokao-Formal* (see Section 3.2). The formal statement that passes the Lean compiler and semantic assessment is then passed to the final proving stage to generate a complete, machine-verifiable Lean proof.

Our approach is, in principle, transferable to other proof assistants such as Isabelle. We focus on Lean 4 due to its broad adoption in recent literature and its well-supported evaluation stack.

## 3.1 *Mathesis-Autoformalizer*: ADVANCING AUTOFORMALIZATION WITH REINFORCEMENT LEARNING

The cornerstone of our pipeline is *Mathesis-Autoformalizer*, a novel model designed to translate informal mathematical problem statements from NL into formal Lean 4 (mathlib Community, 2020) code. Unlike prior approaches that predominantly rely on supervised fine-tuning (SFT) on static datasets, our work presents the first autoformalization model trained with online reinforcement learning via Group Relative Policy Optimization (GRPO) (Shao et al., 2024), using reward signals for syntactic validity and semantic correctness, and incorporating prover-derived feedback. Training proceeds in two stages: GRPO aligns the model to syntactic compilability and semantic faithfulness, after which Direct Preference Optimization (DPO) further aligns it to downstream proof success by preferring formalizations that lead to Lean-verified proofs. We refer to this two-stage procedure as Hierarchical Preference Optimization (HPO).

**Composite Rewards for Autoformalization** Let $\pi_\theta$ represents the translator LLM policy parameterized by $\theta$, and $\pi_{ref}$ be a fixed reference policy (typically the SFT model). In our experiment, the policy $\pi_\theta$ is initialized from Kimina-Autoformalizer (Wang et al., 2025), the previous state-of-the-art 7B autoformalizer trained via supervised fine-tuning with expert iteration to translate natural language problems into Lean 4 code. This initialization provides a strong structural prior, equipping the model with canonical Lean formatting patterns and facilitating subsequent reinforcement learning. For a given natural language input $x \in \mathcal{X}$, the policy $\pi_\theta(\cdot|x)$ generates a group of $G$ candidate formal Lean 4 statements as outputs $\{o_1, ..., o_G\}$. The optimization objective is to adjust $\theta$ to maximize the likelihood of generating higher-reward outputs relative to lower-reward ones within the group, while regularizing against large deviations from the reference policy. We define a composite reward with two binary components. For an given input $x$ and a corresponding candidate formalization $o_i$, we define the *Semantic Correctness Reward* $R_{sem}(x, o_i)$ to indicate whether $o_i$ preserves

the semantic meaning of $x$, as judged by an auxiliary LLM evaluator $J_{sem}$:

$$R_{sem}(x, o_i) = \begin{cases} 1 & \text{if } J_{sem}(x, o_i) \text{ judges "Appropriate"} \\ 0 & \text{otherwise} \end{cases}.$$

We also define the *Syntatic Verification Reward* $R_{ver}(o_i)$ to indicate whether $o_i$ is syntactically correct and type-valid under the Lean 4 verifier ($V_{lean}$), checked up to `:=` by `sorry`:

$$R_{ver}(o_i) = \begin{cases} 1 & \text{if } V_{lean}(o_i) \text{ succeeds} \\ 0 & \text{otherwise} \end{cases}.$$

The final overall reward $r_i$ for an output $o_i$ is computed as a combination of the two components: $r_i = R(x, o_i, o_{ref}) = R_{sem} + R_{ver}$. We then leverage the GRPO objective to update $\pi_\theta(\cdot|x)$. We find that this simple yet effective summation strategy indeed leads to state-of-the-art performance. The rationale for this composite reward is to capture two orthogonal constraints: syntactic validity is a necessary condition—non-verifying candidates are unusable by the prover—while semantic correctness ensures that the formalized statement corresponds to the intended problem to be solved. Although both terms are binary, GRPO optimizes within-group preferences rather than absolute magnitudes; we normalize rewards within each group before forming pairwise comparisons, which makes the objective insensitive to uniform rescaling or simple weighting. We adopt the unweighted sum for simplicity, noting that weighting schemes are a reasonable direction for future work.

**Training Data Curation** To effectively identify samples for training, our data curation process employs topic modeling with BERTopic (Grootendorst, 2022) to the natural language informal statements of problems from a pset (Lin et al., 2025) and our in-house Gaokao dataset. BERTopic was selected for its ability to generate coherent topics by leveraging contextual embeddings and clustering, allowing for effective categorization of problems based on their semantic content; this approach is also significantly faster and more cost-effective than using large language models for the same categorization task. We generated embeddings for each statement, performed dimensionality reduction and clustering, and then mapped the resulting topics to predefined mathematical categories in Section 3.3. The categorized data from two sources were then merged. To optimize Reinforcement Learning training efficiency with GRPO, we employed our base model (pre-RL) to perform rollouts (k=14) on each problem, filtering out those yielding rewards with zero standard deviation across the rollouts. The remaining problems demonstrating reward variance were combined with 8,000 problems randomly sampled from the Lean Workbook (Ying et al., 2024), resulting in a final training dataset of approximately 32k problems.

**Hierarchical Preference Optimization for Theorem Proving** GRPO provides a reward-maximizing initialization, aligning the autoformalizer with local objectives of syntactic correctness and semantic validity. In the context of formal theorem proving from natural language, the auto-formalizer formalizes natural language into formal statements aimed at facilitating successful proof generation by the prover. To enhance this, we further fine-tune the autoformalizer using Direct Preference Optimization (DPO) (Rafailov et al., 2023), where preferences are derived from the global success of the downstream proof generation.

**DPO Training Data Generation** During the data generation phase, for each natural language statement $x$, a group of candidate formal statements $o_i$ are sampled from the autoformalizer $\pi_\theta(\cdot|x)$, where $i$ indexes the candidates. Each $o_i$ undergo syntactic and semantic validation, and those that pass are forwarded to the prover, which attempts to generate proofs $z^i$. Preferences are assigned based on the successful completion of the proof verified by Lean, yielding data tuple $\{x, o_i^w, z_i^w\}$ for successful cases, and $\{x, o_i^l, z_i^l\}$ for failed attempts.

**DPO Training** The training configuration employs a single epoch with a learning rate of $1 \times 10^{-5}$. The KL regularization coefficient $\beta$ is set to 0.1, penalizing deviations from the reference model. Optimization is applied to full parameters, with a warmup ratio of 0.05. To manage memory usage efficiently, training is conducted using DeepSpeed zero3 offload.

DPO fine-tuning enhances alignment with task-grounded outputs, thereby mitigating mismatches between reward function and actual task objectives. Compared to GRPO, DPO is a more sample-efficient and stable alternative that performs offline preference learning and eliminates the need for a separate reward model (Rafailov et al., 2023; Ouyang et al., 2022). However, its effectiveness heavily relies on a strong base model to generate meaningful candidate outputs and better exploit

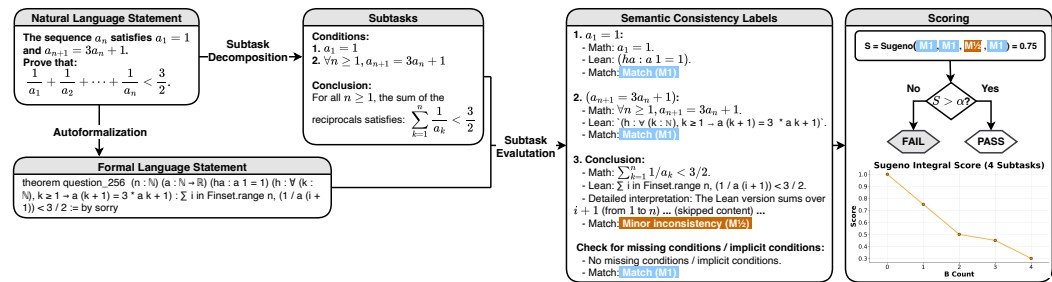

Figure 3: Overview of LeanScorer semantic evaluation framework.

preference signals (Tu et al., 2025; Wang et al., 2024; Pan et al., 2025). To this end, we first apply GRPO to establish a strong initialization before proceeding with DPO fine-tuning.

## 3.2 *LeanScorer*: FINE-GRAINED SEMANTIC EVALUATION FOR AUTOFORMALIZATION

The evaluation of autoformalized problem statements typically follows a two-step process: a syntactic check conducted by the Lean compiler, followed by a semantic check, which verifies that the formal statement preserves the intended meaning of the original natural language input. Existing approaches to semantic correctness (Lin et al., 2025; Gao et al., 2024; Ying et al., 2024) rely on LLMs and are limited to binary judgments (i.e., correct or incorrect). In this section, we propose *LeanScorer*, a novel framework that produces a continuous correctness score, enabling fine-grained and task-adaptive assessment. Figure 3 provides an overview.

**Subtask Decomposition and Evaluation**   Given a natural language (NL) statement and its corresponding formal (FL) statement, *LeanScorer* first decomposes the NL statement into subtasks such as premises and conclusions. Each subtask is then aligned with its counterpart in the FL statement and assigned one of three labels: Match (M1), Minor Inconsistency (M½), or Major Inconsistency (M0). Specifically, clear mathematical inequivalence or omitted conditions are labeled as "Major Inconsistency"; semantically equivalent but divergent expressions are labeled as "Minor Inconsistency"; and exact alignment in both content and structure is labeled as "Match". This three-level labeling scheme enables nuanced semantic evaluation while accommodating the inherent variability in LLM outputs. Any extra assumptions or conditions in the formal statement that do not correspond to a natural-language subtask are marked as misaligned; likewise, any natural-language subtask missing from the formalization is also treated as a misalignment. This enables LeanScorer to detect both additions and omissions. Prompts are provided in Appendix E.

**Aggregation via Sugeno Fuzzy Integral**   To compute an overall correctness score, we aggregate subtask-level labels using the Sugeno Fuzzy Integral (Sugeno, 1974), a well-established method in multi-criteria decision-making (MCDM) (Wieczynski et al., 2024). We design a customized fuzzy measure that tolerates minor inaccuracies in the LLM judgments while enforcing strict criteria: it strictly penalizes any formalization with a subtask labeled as a Major Inconsistency, grants a full score when all subtasks are labeled as Match, and multiple Minor Inconsistencies incur proportional deductions. This method enables robust label aggregation under LLM outputs variability and upholds rigorous evaluation criteria.

Let $N = \{1, 2, ..., n\}$ denote the set of $n$ subtasks, and let $L = \{l_1, ..., l_n\}$ the corresponding label set, where each $l_i \in \{M1, M½, M0\}$ denotes the semantic consistency label of the $i$-th subtask. We define an evaluation mapping function $f : L \to [0, 1]$, where we set $f(M1) = 1.0$, $f(M½) = 0.5$, and $f(M0) = 0$. To aggregate the semantic quality over a subset $s \subseteq L$, we introduce the fuzzy measure $\mu(s)$ defined as:

$$\mu(s) = \begin{cases} 0 & \text{if } \exists\, l \in s, l = M0, \\ \max\left\{\dfrac{n_s}{n} \cdot (1 - \delta \cdot n_{M½}), 0\right\} & \text{otherwise} \end{cases} \quad (1)$$

where $n_s$ and $n_{M½}$ denote the number of elements in $s$ and elements labeled and M½ in $s$, respectively. The coefficient $\delta$ is set to 0.1 when $n_{M½} \leq 1$, and 0.2 otherwise. We conducted a sensitivity analysis by varying the value of $f(M½)$ from 0.1 to 0.9 in increments of 0.1 (see Appendix D), and observed that the performance remains robust: the F1 score remains stable at 0.92 for values between

0.1 and 0.5, and drops slightly to 0.88 for values between 0.6 and 0.9. Thus, we set $f(\text{M}\frac{1}{2}) = 0.5$ in all experiments. To further validate our design choices, we also conduct an ablation study comparing Sugeno Fuzzy Integral–based aggregation against alternative aggregation methods in Appendix D.

Next, let the labels be sorted in ascending order as $f(l_{\pi(1)}) \leq f(l_{\pi(2)}) \leq \cdots \leq f(l_{\pi(n)})$, with $\{\pi(1), \pi(2), \ldots, \pi(n)\}$ representing the corresponding indices. For each $i \in \{1, \ldots, n\}$, we define the suffix set $s_i = \{l_{\pi(i)}, \ldots, l_{\pi(n)}\}$, which represents the subset consisting of $l_{\pi(i)}$ and all labels ranked after it. The overall LeanScore is then computed as:

$$S(L, f, \mu) = \max_{1 \leq i \leq n} \min \left( f(l_{\pi(i)}), \ \mu(s_i) \right). \tag{2}$$

A decision threshold $\alpha \in [0, 1]$ may optionally be applied to map the LeanScore to a binary decision.

### 3.3 THE GAOKAO-FORMAL BENCHMARK

To advance automatic formal theorem proving from natural language, we introduce the *Gaokao-Formal* benchmark. Unlike existing benchmarks that focus primarily on formal-to-formal proving or sometimes exclude those problems that are hard to formalize, *Gaokao-Formal* specifically targets the difficulties of auto-formalizing diverse and complex natural language mathematical statements, aiming to motivate real-world applications of formal reasoning.

This benchmark consists of 495 proof problems from China's National Higher Education Entrance Examination (Gaokao, 2008-2025), often include sub-questions. Each instance contains the original Chinese problem

Table 1: Summary of Gaokao-Formal Benchmark Categories.

| Category | Count |
|---|---|
| Functions | 168 |
| Sequences & Series | 148 |
| Analytic Geometry | 76 |
| Comprehensive Questions | 49 |
| Inequality | 28 |
| Trigonometry | 22 |
| Probability & Combinatorics | 4 |

statement, an English translation, and a human-expert formalized Lean 4 formal statement. A summary of question categories is provided in Table 1, with example problems available in Appendix C.

**Remark.** The Gaokao dataset utilized in this study consists of publicly available official statistics, administered by government authorities. It is classified as government-managed public information and does not involve privately copyrighted material.

## 4 EXPERIMENTS

We evaluate our proposed approach along three axes. First, we assess the semantic consistency evaluation framework LeanScorer. Second, we compare the Mathesis-Autoformalizer with state-of-the-art autoformalization baselines. Finally, we measure how improved autoformalization affects formal theorem proving from natural language. Experiments are conducted on our newly introduced Gaokao-Formal benchmark and the widely adopted MiniF2F-test set. We compare our Mathesis-Autoformalizer with strong API-based and open-source baselines, including the prior state of the art Kimina-Autoformalizer (Wang et al., 2025) and Herald-Autoformalizer Gao et al. (2025). For pipeline evaluation, we pair each autoformalizer with the current best-in-class 7B provers—Kimina-Prover (Wang et al., 2025), Goedel-Prover (Lin et al., 2025), and DeepSeek-Prover-V2 (Ren et al., 2025) with COT mode—tasking the prover to generate proofs from the formal statements produced by the respective autoformalizer and reporting proof success rate.

Autoformalization quality is measured by two standard metrics: Lean Check success rate at k candidates (LC@k) for solely syntactic validity, and Lean Check combined with LeanScorer semantic checking (LC+LSC@k) for overall correctness, jointly assessing syntactic and semantic correctness. We reported results for $k = 1$ and $k = 6$. Mathesis pipeline performance is measured as the proof success rate, given a fixed search budget of 32 attempts per problem, by convention. Additional implementation details, including training configurations and prompts, are provided in the Appendix.

### 4.1 SEMANTIC CONSISTENCY EVALUATION FRAMEWORK

We first validate our LeanScorer against two widely used but indirect, ground-truth–free baselines: LLM-as-a-Judge (Lin et al., 2025; Gao et al., 2024) and Re-informalization (Ying et al., 2024). The

evaluation uses a human-annotated subset of Gaokao-Formal containing both correct and incorrect autoformalizations. Annotation details are in Appendix G. Unless otherwise noted, the decision threshold is set to $\alpha = 0.6$. LLM-as-a-Judge directly prompts an LLM to decide whether a formal statement matches a natural-language statement and outputs a single semantic-correctness judgment. Re-informalization evaluates semantic consistency by first back-translating the formal statement into natural language and then comparing it with the original input, involving two LLM calls per round. Both baselines incorporate test-time scaling and accept a formalization only when all four outputs agree. In contrast, LeanScorer performs LLM-assisted subtask decomposition and consistency annotation in a single pass using two LLM calls, one for decomposition and one for annotation, followed by a computationally efficient aggregation step to produce a semantic-consistency score.

Table 2 shows that LeanScorer achieves an F1 score of 92%, significantly outperforming both the LLM-as-a-Judge (85% F1) and Re-informalization (65% F1). While LLM-as-a-Judge has 100% recall, its 73% precision is low, yielding many false positives. Re-informalization is the opposite, with 93% precision but 50% recall. LeanScorer delivers both high precision

Table 2: Semantic evaluation framework performance measured by agreement with human annotations.

| Method | Precision | Recall | F1 |
|---|---|---|---|
| LLM-as-a-Judge | 73 | **100** | 0.85 |
| Re-informalization | 93 | 50 | 0.65 |
| **LeanScorer (Ours)** | **94** | 89 | **0.92** |

(94%) and high recall (89%), indicating a balanced and reliable semantic consistency checker for formalization evaluation. This balance is important for downstream performance: high precision ensures that only semantically correct formalizations reach the prover, while high recall increases coverage but also admits more misaligned statements. Since we require a Lean-verified proof for the original natural-language problem, maintaining both high precision and high recall is essential for reliable overall performance.

## 4.2 Autoformalization from Natural Language

The core contribution of our work is the *Mathesis-Autoformalizer*, designed to translate natural language problems into syntactically valid and semantically faithful Lean4 statements. In this section, we evaluate two variants: Mathesis-Autoformalizer, obtained after the GRPO stage, and Mathesis-Autoformalizer-HPO, obtained after the full two-stage Hierarchical Preference Optimization training (GRPO followed by DPO). These models are compared against state-of-the-art API-based and open-source baselines, including Kimina-Autoformalizer, which serves as our base model and was the strongest publicly available 7B autoformalizer before our work. Evaluations are conducted on the MiniF2F-test, Putnam, and Gaokao-Formal benchmarks. Performance is measured using Lean Check (LC@k) for syntactic validity and Lean Check plus LeanScorer Semantic Check (LC+LSC@k) for overall correctness. For fair comparison, our models are compared directly against open-source 7B baselines, with top scores underlined to highlight improvements at equal model size. API models, which are much larger in size, are compared within the group, with the top scores shown in bold. All models are evaluated with a 600-second timeout per prompt.

Table 3 presents the results. Both Mathesis variants outperform all baselines across datasets and budgets, with the largest gains on the more challenging Gaokao-Formal benchmark. Specifically, Mathesis-Autoformalizer-HPO achieves an LC+LSC@6 score of 71% on Gaokao-Formal, surpassing the previous best of 49% by Kimina-Autoformalizer with an absolute improvement of 22 percentage points and a 45% relative gain. On another challenging dataset Putnam, Mathesis-HPO achieves 30% LC+LSC@6 score, compared with 10% from Kimina-Autoformalizer and 9% from Herald-Autoformalize, corresponding to 200% and 233% relative gains, respectively. On MiniF2F-test, it reaches a new record LC+LSC score of 96%. Mathesis-HPO reaches 96% LC+LSC@6, establishing a new state of the art for overall pass rate on this dataset. Mathesis-HPO constantly outperforms all much larger API-based models across datasets in both LC@6 and LC+LSC@6, with an average of 311% and 86%, resp., except for DeepSeek-R1 on challenging Putnam in LSC. In terms of LC, our method improves over Kimina-Autoformalizer by 4 points on MiniF2F and 9 points on Gaokao-Formal, demonstrating superior syntactic reliability grounded in Lean's compiler feedback.

These improvements support the effectiveness of our training methodology. By employing reinforcement learning with a composite reward that combines Lean-based syntactic signals and

Table 3: Quality assessment of formalized statements. LC denotes Lean Check; LSC denotes LeanScorer Semantic Check. Top scores for API and open-source models are **bold** and underlined.

| Model | k | MiniF2F-Test | | Putnam | | Gaokao-Formal | |
|---|---|---|---|---|---|---|---|
| | | LC | LC+LSC | LC | LC+LSC | LC | LC+LSC |
| *API Models* | | | | | | | |
| o3-mini | 1 | 58 | 45 | 7 | 3 | 38 | 25 |
| | 6 | 87 | 77 | 13 | 7 | 70 | 54 |
| GPT-4o | 1 | 50 | 36 | 9 | 3 | 20 | 13 |
| | 6 | 80 | 65 | 24 | 9 | 48 | 28 |
| Doubao-1.5 | 1 | 48 | 40 | 7 | 4 | 19 | 15 |
| | 6 | 77 | 70 | 16 | 10 | 45 | 32 |
| Gemini-2.0 | 1 | 56 | 41 | 20 | 10 | 36 | 22 |
| | 6 | 80 | 71 | 42 | 26 | 66 | 47 |
| Deepseek-V3 | 1 | 76 | 61 | 10 | 4 | 54 | 36 |
| | 6 | **91** | **84** | 22 | 10 | 69 | 56 |
| Deepseek-R1 | 1 | 54 | 44 | 29 | 14 | 45 | 30 |
| | 6 | 86 | 76 | **58** | **37** | **81** | **57** |
| *Open-Source Models* | | | | | | | |
| Herald-Autoformalizer | 1 | 80 | 41 | 35 | 4 | 56 | 14 |
| | 6 | 95 | 69 | 64 | 9 | 78 | 27 |
| Kimina-Autoformalizer | 1 | 83 | 61 | 7 | 2 | 50 | 21 |
| | 6 | 100 | 91 | 30 | 10 | 91 | 49 |
| **Mathesis-Autoformalizer** | 1 | 92 | 69 | 31 | 9 | 88 | 45 |
| | 6 | 100 | 95 | 65 | 25 | 98 | 67 |
| **Mathesis-Autoformalizer-HPO** | 1 | 99 | 79 | 38 | 10 | 93 | 50 |
| | 6 | 100 | 96 | 73 | 30 | 98 | 71 |

LeanScorer-based semantic signals, the model learns to produce formalizations that are both syntactically sound and semantically faithful—an essential capability for reliable automated reasoning.

## 4.3 AUTOFORMALIZATION-DRIVEN THEOREM PROVING

The ultimate measure of our approach's effectiveness is its performance on formal theorem proving from natural language. We posit that the quality of the autoformalization critically influences the final proving accuracy. Accordingly, we pair each autoformalizer with downstream provers and evaluate the proof accuracy on MiniF2F-Test and Gaokao-Formal. The results are shown in Figure 4.

**Autoformalization quality correlates with proving accuracy.** There is a strong positive correlation between autoformalization quality and proving accuracy, where proving accuracy denotes the rate at which the prover produces complete and correct formal proofs. Across all provers and both datasets, accuracy improves consistently as the autoformalizer improves. For any fixed prover, replacing Herald or Kimina with our Mathesis-HPO yields substantial gains. On Gaokao-Formal, replacing Herald with Mathesis-HPO improves the accuracy of Goedel-prover, Kimina-prover, and Deepseek-Prover-V2 by 86%, 116%, and 122%, respectively. relative to Kimina, the gains are 49%, 33%, and 42%. On MiniF2F-Test, the same replacement improves the three provers by 98%, 77%, and 51% over Herald, and by 6%, 9%, and 5% over Kimina.

These improvements arise because a stronger autoformalizer supplies the prover with a larger number of compiler-valid and semantically faithful Lean statements, providing well-structured and solvable goals. Mathesis-Autoformalizer-HPO further outperforms Mathesis-Autoformalizer because the DPO stage incorporates prover-derived preferences, favoring formalizations that not only compile and align semantically but also lead to successful proof completion. Additional evaluations on autoformalizer output before and after DPO are provided in Appendix H.

**Autoformalizer improvements dominate prover upgrades.** For problems that are intrinsically harder to formalize, the performance gains from a stronger autoformalizer often exceed those from a stronger prover. On Gaokao-Formal, holding the prover fixed and replacing Herald with Mathesis-HPO increases accuracy by 86% for Goedel-Prover, 116% for Kimina-Prover, and 122% for DeepSeek-Prover-V2. By contrast, with the autoformalizer fixed, upgrading the prover from

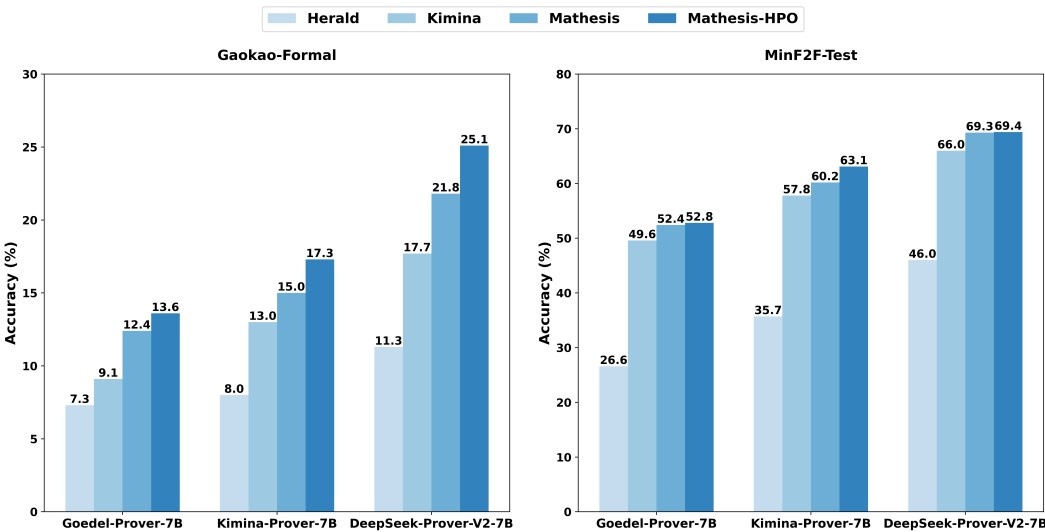

Figure 4: Performance of formal theorem proving from natural language. In each subfigure, bars are grouped by prover on the x-axis, and the y-axis reports pass@32 proving accuracy. Bar colors encode the autoformalizers Herald-Autoformalizer (Herald), Kimina-Autoformalizer (Kimina), Mathesis-Autoformalizer (Mathesis), and Mathesis-Autoformalizer-HPO (Mathesis-HPO).

Geodel-Prover to Deepseekp-Prover-V2 increases accuracy by 55% under Herald and 95% under Kimina. This disparity underscores a fundamental challenge in real-world mathematical reasoning: before a proof can be constructed, natural language problems must first be accurately converted into formal statements that theorem provers can process. Suboptimal formalizations, such as encoding problems using difficult-to-unfold built-in functions or embedding questions within overly complex mathematical definitions, create significant obstacles for downstream proof search (this phenomenon is also observed by Lin et al. (2025)). Our results indicate that formalization quality constitutes a critical bottleneck, as provers are hindered by suboptimal formalizations that obscure the problem's structure or introduce unnecessary complexity. Hence, achieving high accuracy requires both a strong autoformalizer and a strong prover, attesting to the critical role of formalization.

**Ablations on pipeline design.** As shown in Table 3 and Figure 4, adding HPO on top of GRPO improves pass rate from 67% to 71% on Gaokao-Formal and from 95% to 96% on MiniF2F. In downstream proving, Mathesis-HPO consistently outperforms Mathesis cross provers by an average of 13% on Gaokao-Formal and 2% on MiniF2F. These results show that HPO's prover-derived feedback as reward yields measurable end-task gains and better aligns the autoformalizer with proof success beyond gains attributable to syntax and semantic checks alone.

## 5 LIMITATION

While our approach demonstrates significant improvements, formal theorem proving from natural language remains far from full automation and real-world deployment. During the development of Mathesis-Autoformalizer and LeanScorer, we discovered that the correctness of formalization is often difficult to define directly and must be determined based on whether it affects the proof process (e.g., whether functions need explicit domain declarations). This ambiguity in formalization correctness presents ongoing challenges for both training and semantic evaluation.

## 6 CONCLUSION

Motivated by the goal of bringing formal reasoning to real-world proof questions, we introduce the task of formal theorem proving from natural language and present Mathesis as a comprehensive solution. Our Mathesis-Autoformalizer, trained through online reinforcement learning with novel Hierarchical Preference Optimization, and LeanScorer for robust semantic evaluation both achieve significant improvements over existing methods. This work lays a solid foundation for future advances toward fully integrated and scalable formal reasoning systems, bringing formal theorem proving to real-world mathematical problem solving.

ETHICS STATEMENT

The research work presented in this paper adheres strictly to the ICLR Code of Ethics. This study does not involve human subjects, and there are no potential conflicts of interest or fairness concerns related to the work.

The Gaokao-Formal Benchmark introduced in this paper is derived from publicly available official statistics administered by government authorities. As government-managed public information, the dataset is free from private copyright restrictions. The models, code, and data released in this work are intended solely for academic and research purposes. They do not pose privacy or security risks and comply with legal and ethical standards.

We confirm that we have read and complied with the ICLR Code of Ethics.

REPRODUCIBILITY STATEMENT

To ensure the reproducibility of our research, we provide detailed information about our methodology, datasets, and experimental setup. The source code for our project, including the implementation of the Mathesis-Autoformalizer and the LeanScorer evaluation framework, is available at `https://github.com/Huawei-AI4Math/Mathesis`.

The newly introduced Gaokao-Formal benchmark is described in Section 3.3, with further details and examples provided in Appendix C. All experiments were conducted on this benchmark and the publicly available MiniF2F-test set. Our experimental setup, including the baselines used for comparison, evaluation metrics, and results, is detailed in Section 4.

The training configurations and hyperparameters for the Mathesis-Autoformalizer are provided in Appendix A. This includes details on the Group Relative Policy Optimization (GRPO) and Hierarchical Preference Optimization (HPO) stages. The prompts used for all language models in our experiments, including the autoformalization prompts and the prompts for the LLM-as-a-Judge and LeanScorer frameworks, are available in Appendix E.

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

APPENDIX

## A    TRAINING DETAILS FOR *Mathesis-Autoformalizer*

The training of our *Mathesis-Autoformalizer* model, which employs Group Relative Policy Optimization (GRPO), involves several key hyperparameters and implementation choices as briefly mentioned in Section 3.1. The policy $\pi_\theta$ is initialized from Kimina-Autoformalizer (Wang et al., 2025). We employ Parameter-Efficient Fine-Tuning (PEFT) via Low-Rank Adaptation (LoRA) (Hu et al., 2022), configured with a rank $r = 16$ and $\alpha = 32$. LoRA is applied to the attention projection layers of the base model. The optimization is performed using the AdamW optimizer (Loshchilov & Hutter, 2019) with a learning rate of $1 \times 10^{-6}$ and gradient checkpointing to manage memory usage.

For the GRPO algorithm itself, we sample $G = 14$ candidate formal statements per input natural language problem $x$. The Kullback-Leibler (KL) divergence coefficient $\beta$, which regularizes the policy updates against the reference SFT policy, is set to $0.04$. The policy model $\pi_\theta$ is updated once per sampling/exploration phase (i.e., $\mu = 1$, meaning updates occur after each group of $G$ generations for a given input $x$ is processed and rewarded). To enhance efficiency, reward computations (syntactic verification via Lean and semantic assessment) are parallelized using Python's `asyncio` library. The overall training pipeline is managed using the Hugging Face `transformers` (Wolf et al., 2020) and `trl` (von Werra et al., 2020-2024) libraries. Experiment progress and results are logged using Weights & Biases (Biewald, 2020).

## B    DATA DEDUPLICATION AND CONTAMINATION ANALYSIS

To ensure that the performance gains reported in this paper reflect genuine reasoning capabilities rather than memorization, we conducted a rigorous data contamination analysis. This section details our methodology and presents the overlap statistics of our main evaluation benchmark, GAOKAO-FORMAL, as well as standard benchmarks MiniF2F and PutnamBench against our three primary training data sources: the In-house Gaokao Corpus (Combined), Goedel P-Set, and Lean Workbook.

### B.1    METHODOLOGY

We implemented a strict lexical overlap detection pipeline to audit potential data leakage. The process consists of the following steps:

1. **Normalization:** All text data from both training and evaluation sets was normalized using NFKC normalization, converted to lowercase, and had whitespace collapsed.
2. **N-gram Extraction:** We extracted all 50-character substrings (windows) starting at word boundaries from the evaluation benchmarks.
3. **Matching:** An Aho-Corasick automaton was constructed to stream the training corpora and detect matches efficiently.

For each problem statement $i$ in the evaluation set, we calculated an overlap ratio $\eta_i$, defined as:

$$\eta_i = \frac{\text{matched windows}}{\text{total windows}} \tag{3}$$

Based on this ratio, problems were categorized into three levels of contamination:

- **Clean:** $\eta_i < 0.2$ (Less than 20% overlap)
- **Suspicious:** $0.2 \leq \eta_i < 0.8$ (Between 20% and 80% overlap)
- **Dirty:** $\eta_i \geq 0.8$ (Greater than 80% overlap, indicating near-duplicates)

### B.2    RESULTS

We performed the analysis for GAOKAO-FORMAL, MiniF2F-test, and PutnamBench against three distinct training subsets.

**Analysis 1: Overlap against In-house Gaokao Corpus.** Table 4 shows the contamination rates against our primary In-house Gaokao training corpus (Combined English Informal). GAOKAO-FORMAL shows negligible overlap (1.2% suspicious, 0% dirty). Similarly, MiniF2F and Putnam-Bench are entirely clean relative to this training source.

Table 4: Contamination Analysis against **In-house Gaokao Corpus (Combined)**

| Evaluation Set | Total Entries | Clean ($< 0.2$) | Suspicious ($[0.2, 0.8)$) | Dirty ($\geq 0.8$) |
|---|---|---|---|---|
| Gaokao-Formal | 495 | 489 (98.8%) | 6 (1.2%) | **0 (0.0%)** |
| MiniF2F | 488 | 488 (100%) | 0 (0.0%) | **0 (0.0%)** |
| PutnamBench | 661 | 661 (100%) | 0 (0.0%) | **0 (0.0%)** |

**Analysis 2: Overlap against Goedel P-Set.** Table 5 presents the results against the Goedel P-Set. While GAOKAO-FORMAL remains robust with zero dirty matches, existing benchmarks show significant contamination. Specifically, MiniF2F contains 31 dirty proofs (6.4%) and PutnamBench contains 41 dirty proofs (6.2%), suggesting that models trained on the P-Set may memorize solutions for these standard benchmarks.

Table 5: Contamination Analysis against **Goedel P-Set**

| Evaluation Set | Total Entries | Clean ($< 0.2$) | Suspicious ($[0.2, 0.8)$) | Dirty ($\geq 0.8$) |
|---|---|---|---|---|
| Gaokao-Formal | 495 | 479 (96.8%) | 16 (3.2%) | **0 (0.0%)** |
| MiniF2F | 488 | 372 (76.2%) | 85 (17.4%) | **31 (6.4%)** |
| PutnamBench | 661 | 433 (65.5%) | 187 (28.3%) | **41 (6.2%)** |

**Analysis 3: Overlap against Lean Workbook.** Table 6 displays the overlap against the Lean Workbook dataset. All three evaluation benchmarks are virtually free of contamination from this source, with GAOKAO-FORMAL showing 100% clean entries.

Table 6: Contamination Analysis against **Lean Workbook**

| Evaluation Set | Total Entries | Clean ($< 0.2$) | Suspicious ($[0.2, 0.8)$) | Dirty ($\geq 0.8$) |
|---|---|---|---|---|
| Gaokao-Formal | 495 | 495 (100%) | 0 (0.0%) | **0 (0.0%)** |
| MiniF2F | 488 | 486 (99.6%) | 2 (0.4%) | **0 (0.0%)** |
| PutnamBench | 661 | 660 (99.8%) | 1 (0.2%) | **0 (0.0%)** |

In conclusion, our analysis confirms that GAOKAO-FORMAL is not contaminated by any of the training datasets used in this work. Furthermore, the detection of "Dirty" samples in MiniF2F and PutnamBench against the Goedel P-Set highlights the importance of using fresh, uncontaminated benchmarks like GAOKAO-FORMAL to accurately assess generalization in formal theorem proving.

### B.3 CONTAMINATION ANALYSIS AGAINST PRETRAINING CORPORA

To further ensure that the performance on **Gaokao-Formal** reflects genuine reasoning capabilities rather than memorization of pretraining data, we conducted an extensive contamination analysis against major open-source pretraining corpora. Specifically, we checked for N-gram overlap against **The Pile** (Pile-train), **DCLM-baseline**, and five snapshots of **CommonCrawl** (CC-2025-05 through CC-2025-21).

As shown in Table 7, **Gaokao-Formal** exhibits **0.0%** "Dirty" matches (defined as $\geq 80\%$ 50-char N-gram overlap) across all evaluated pretraining datasets. This confirms that the benchmark problems were not seen during the pretraining phase of standard base models.

## C DETAILS OF GAOKAO-FORMAL BENCHMARK

**Problem-Type Diversity** Unlike benchmarks that may filter out problem types with less developed theorem libraries (e.g., geometry, combinatorics), *Gaokao-Formal* includes all such problems

| Benchmark | Pile-train Dirty (%) | DCLM-base Dirty (%) | CC-25-05 Dirty (%) | CC-25-08 Dirty (%) | CC-25-13 Dirty (%) | CC-25-18 Dirty (%) | CC-25-21 Dirty (%) |
|---|---|---|---|---|---|---|---|
| **Gaokao-Formal** | **0.0** | **0.0** | **0.0** | **0.0** | **0.0** | **0.0** | **0.0** |
| *Reference Benchmarks:* | | | | | | | |
| AIME 2025 | 0.0 | 0.0 | 0.0 | 0.0 | 0.0 | N/A | N/A |
| AMC 2023 | 0.0 | 0.0 | 0.0 | 0.0 | 0.0 | 0.0 | 0.0 |
| TruthfulQA | 0.1 | 0.1 | 1.0 | N/A | N/A | N/A | N/A |

Table 7: Contamination analysis of **Gaokao-Formal** against massive pretraining corpora. "Dirty" indicates the percentage of samples with $\geq 80\%$ overlap. Dash (–) indicates no data available for that snapshot. Gaokao-Formal remains entirely clean across all sources.

---

**MiniF2F**

**NL:** If $x$ and $y$ are positive integers for which $2^x 3^y = 1296$, prove that $x + y = 8$.

**FL:** `theorem amc12b_2004_p3` $(x\ y : \mathbb{N})$ $(h_0 : 2\ \hat{}\ x * 3\ \hat{}\ y = 1296)$ : x + y = 8 := by sorry

**Gaokao-Formal**

**NL:** Let $m$ be a positive integer, and let $a_1, a_2, \cdots, a_{4m+2}$ be an arithmetic sequence with a non-zero common difference. If two terms $a_i$ and $a_j$ $(i < j)$ are removed from the sequence such that the remaining $4m$ terms can be evenly divided into $m$ groups, and each group of 4 numbers forms an arithmetic sequence, then the sequence $a_1, a_2, \cdots, a_{4m+2}$ is called an $(i, j)-$separable sequence. For $m \geq 3$, prove that the sequence $a_1, a_2, \cdots, a_{4m+2}$ is a $(2, 13)-$separable sequence.

**FL:** `theorem gaokaoformal_g4` $(m : \mathbb{N})$ $(hm : 1 \leq m)$ $(a : \mathbb{N} \to \mathbb{R})$ $(ha : \exists (d:\mathbb{R}), d \neq 0 \wedge (\forall (n:\mathbb{N}), (n \geq 1 \wedge n \leq 4*m+1) \to a(n+1) = a\ n + d))$ $(sep : (\mathbb{N} \times \mathbb{N}) \to \text{Prop})$ $(h\_sep : \forall (i\ j:\mathbb{N}), (i \geq 1 \wedge i \dot{\jmath} \wedge j \leq 4*m+2) \to sep(i,j) = (\exists (f : \mathbb{N} \to \mathbb{N}), (\forall (h:\mathbb{N}), (h \geq 1 \wedge h \leq 4*m+2 \wedge h \neq i \wedge h \neq j) \to (f\ h \geq 1 \wedge f\ h \leq m)) \wedge (\forall (g:\mathbb{N}), \text{let } S := \{h:\mathbb{N} \mid h \geq 1 \wedge h \leq 4*m+2 \wedge h \neq i \wedge h \neq j \wedge f\ h = g\};$ $(g \geq 1 \wedge g \leq m) \to (\text{Nat.card } S = 4 \wedge (\exists (p: \mathbb{N} \to S), (\forall (k\ l:\mathbb{N}), (k \geq 1 \wedge k \leq 4 \wedge l \geq 1 \wedge l \leq 4 \wedge k \neq l) \to p\ k \neq p\ l) \wedge (\exists (d':\mathbb{R}), \forall (k:\mathbb{N}), (k \geq 1 \wedge k \leq 3) \to a(p(k+1)) = a(p\ k) + d')))))) : m \geq 3 \to sep(2,13) :=$ by sorry

---

Figure 5: Comparison of the complexity of the problems in MiniF2F v.s. Gaokao-Formal

as they appear in the Gaokao exams. This encourages broader model capabilities and contributes to the expansion of Lean 4's Mathlib (mathlib Community, 2020).

**Autoformalization Complexity**  Many existing benchmarks simplify or exclude problems where the primary challenge lies in the formalization. *Gaokao-Formal* retains these, especially in its "comprehensive questions" category, which features problems with multi-domain concepts, novel definitions within question, or complex linguistic structures, thereby rigorously testing LLM abstraction capabilities. We provide an example of this kind of question, comparing it with one MiniF2F question in Figure 5.

**Remark on Copyright Status:**  The Gaokao dataset utilized in this study consists of publicly available official statistics, administered by government authorities. It is classified as government-managed public information and does not involve privately copyrighted material.

# D  AGGREGATION DESIGN AND SENSITIVITY ANALYSIS OF THE LEANSCORER

## D.1  EVALUATION AND ABLATION OF AGGREGATION METHODS FOR LEANSCORER

To demonstrate the superiority of our Sugeno integral-based scoring framework, we conduct comprehensive ablation studies comparing our design against four alternative aggregation strategies:

**Binary Aggregation**: A strict one-vote-veto scheme, where the presence of any minor or major inconsistency results in a score of 0, and only all-M1 evaluations receive a score of 1.0:

$$S(L) = \begin{cases} 0 & \exists l_i \in \{\text{M½}, \text{M0}\} \\ 1 & \text{otherwise} \end{cases}$$

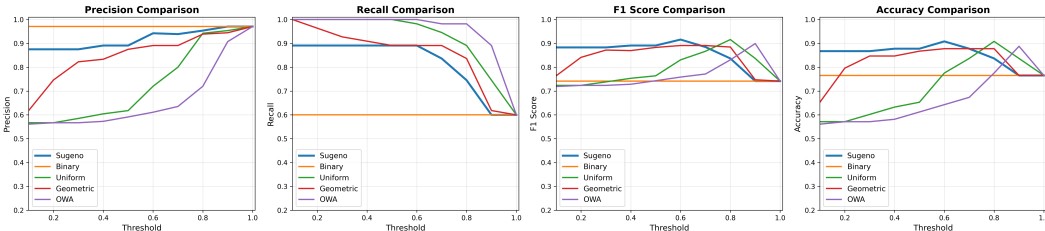

Figure 6: Performance comparison of aggregation methods Sugeno, Binary, Uniform, Geometric, and OWA across thresholds ranging from 0.1 to 1.0.

Table 8: Comparison of aggregation methods, reporting best threshold, Precision, Recall, F1, Accuracy, and Average Accuracy ($\pm$ standard deviation)

| Methods | Best Threshold | Precision | Recall | F1 | Accuracy | Average Acc |
|---|---|---|---|---|---|---|
| Sugeno (ours) | 0.6 | 0.94 | 0.89 | 0.92 | 0.91 | 0.85($\pm$0.046) |
| Binary | N/A | 0.97 | 0.6 | 0.74 | 0.77 | 0.77($\pm$ 0.0) |
| Uniform | 0.8 | 0.94 | 0.89 | 0.92 | 0.91 | 0.72($\pm$ 0.117) |
| Geometric | 0.6 | 0.89 | 0.89 | 0.89 | 0.87 | 0.82 ($\pm$ 0.069) |
| OWA | 0.9 | 0.91 | 0.89 | 0.90 | 0.89 | 0.66 ($\pm$ 0.105) |

**Uniform Averaging**: A straightforward arithmetic mean that assigns equal weight to all subtask evaluations:

$$S(L, f) = \frac{1}{n} \sum_{i=1}^{n} f(l_i)$$

with the mapping function $f(\text{M1}) = 1.0$, $f(\text{M½}) = 0.5$, and $f(\text{M0}) = 0$.

**Geometric Mean**: An aggregation method more sensitive to low scores, computing the geometric mean of all evaluation scores. To avoid a zero product, M0 is mapped to 0.01.

$$S(L, f) = \left( \prod_{i=1}^{n} f(l_i) \right)^{1/n}$$

with the mapping function $f(\text{M1}) = 1.0$, $f(\text{M½}) = 0.5$, and $f(\text{M0}) = 0.01$.

**Ordered Weighted Averaging (OWA)**: A position-weighted approach that emphasizes higher-ranked evaluations, with weights decreasing linearly. Here, we adopt a descending order $f(l_{\pi(1)}) \geq f(l_{\pi(2)}) \geq \cdots \geq f(l_{\pi(n)})$:

$$S(L, f) = \sum_{i=1}^{n} w_i \cdot f(l_{\pi(i)})$$

where $w_i = \frac{n-i+1}{\sum_{j=1}^{n} j}$, $\sum_{i=1}^{n} w_i = 1$, and the mapping function is $f(\text{M1}) = 1.0$, $f(\text{M½}) = 0.5$, and $f(\text{M0}) = 0$.

Figure 6 presents the performance curves of all scoring methods across thresholds ranging from 0.1 to 1.0, while Table 8 summarizes each method's optimal performance and stability, reporting both mean and standard deviation of accuracy across thresholds. Our Sugeno integral achieves the best overall performance, with an F1 score of 0.92 and accuracy of 0.91 at the threshold of 0.6. It also exhibits the highest mean accuracy and the lowest standard deviation, highlighting both its effectiveness and robustness.

Among the methods, the Binary method achieves the highest precision of 0.97 but suffers from a very low recall of 0.6, which reduces its F1 score to 0.74. The Uniform averaging method attains a competitive F1 score of 0.92 and accuracy of 0.91, but its performance varies substantially with the threshold, reflected by a mean accuracy of 0.72 and a standard deviation of 0.117, which, in contrast to the more stable performance of the Sugeno method (accuracy mean = 0.85, std = 0.046). The Geometric mean demonstrates moderate stability but does not surpass the Sugeno integral in

the best F1 or accuracy, whereas OWA achieves strong F1 and accuracy at the optimal threshold but exhibits the poorest stability, with a mean accuracy of 0.66 and a standard deviation of 0.105.

Our Sugeno-based aggregation method offers two key advantages: First, it provides a strong balance by maintaining consistently high precision of 0.94 and recall of 0.89. It avoids the over-rejection characteristic of Binary aggregation and achieves the best F1 and accuracy among all methods, reaching 0.92 and 0.91, respectively. Second, it demonstrates robustness and operational flexibility. The method attains the highest mean accuracy of 0.85 and the lowest standard deviation of 0.046, highlighting its superior stability relative to all baselines. Moreover, as illustrated in Figure 6, it remains effective across a wide threshold range from 0.1 to 0.7, enabling flexible threshold selection and reducing the need for extensive tuning during deployment.

### D.2 SENSITIVITY ANALYSIS OF LEANSCORER

We conduct a sensitivity analysis to assess the impact of the evaluation mapping value for partially correct subtasks, $f(\text{M½})$, on semantic correctness checking performance. As shown in Table 9, we vary $f(\text{M½}) \in \{0.1, 0.2, ..., 0.9\}$, while keeping $f(\text{M1}) = 1.0$ and $f(\text{M0}) = 0$ fixed. We observe that the F1 score remains stable at 0.92 for $f(\text{M½})$ values in the range $[0.1, 0.5]$, indicating robustness to the exact scaling of partial credit. As the value increases beyond 0.5, the F1 score shows a slight degradation, dropping to 0.88 when $f(\text{M½}) \in [0.6, 0.9]$. These results suggest that our metric is relatively robust to the choice of partial reward, we set $f(\text{M½}) = 0.5$ in all experiments.

Table 9: Sensitivity of F1 Score to the Value of $f(\text{M½})$

| $f(\text{M½})$ | 0.1 | 0.2 | 0.3 | 0.4 | 0.5 | 0.6 | 0.7 | 0.8 | 0.9 |
|---|---|---|---|---|---|---|---|---|---|
| F1 Score | 0.92 | 0.92 | 0.92 | 0.92 | 0.92 | 0.88 | 0.88 | 0.88 | 0.88 |

The robustness of LeanScore to the value of $f(\text{M½})$ arises from two factors. First, its max-min aggregation over sorted prefix subsets means small changes to $f(\text{M½})$ rarely affect which subset yields the maximum, unless the changes significantly alter the ordering. Second, the fuzzy measure $\mu(s)$ assigns zero to any set containing an M0 label, making LeanScore more sensitive to fully incorrect outputs than to partial ones. Nevertheless, the partial credit remains critical—by distinguishing M½ from M0, LeanScore captures finer-grained differences in output quality, especially in borderline cases, which would otherwise be treated the same if both were mapped to 0.

We also conduct a sensitivity analysis on the parameter $\delta$ used in fuzzy measure $\mu(s)$, and observe that the F1 score is not sensitive to the choice of $\delta$.

## E PROMPT TEMPLATES

---

**Prompt for Autoformalization (used by all baseline models except Herald)**

You are an expert in formal mathematics. Your task is to translate the given natural language mathematical statement into a formal Lean 4 theorem.
**[Natural language statement]:**
{statement}
Please convert this statement into a precise formal Lean 4 theorem. Follow these guidelines:

1. Start with `theorem` followed by a unique name or the provided ID if available

2. Define the types of all variables (e.g., `a : ℝ` for real numbers)

3. Use appropriate mathematical symbols and notation

4. End with `:= by sorry` to indicate the proof will be completed later

5. Your formalization must exactly capture the mathematical meaning of the statement

**Formal Lean 4 theorem:**

---

## Prompt for LLM-as-a-Judge Semantic Check

You will receive a natural language math problem statement, along with its formal statement in LEAN 4 and, in some cases, a description of mathematical terms. Please evaluate whether the formal LEAN statement appropriately translates the natural language statement based on the following criteria. They are considered different if any of the criteria are not satisfied.

1. **Key Elements**: The fundamental mathematical components, including variables, constants, operations, domain, and codomain are correctly represented in LEAN code.

2. **Mathematical Accuracy**: The mathematical relationships and expressions should be interpreted consistently during translation.

3. **Structural Fidelity**: The translation aligns closely with the original problem, maintaining its structure and purpose.

4. **Comprehensiveness**: All conditions, constraints, and objectives stated in the natural language statement are mathematically included in the LEAN translation.

When doing evaluation, break down each problem statement into components, match the components, and evaluate their equivalence. Think step-by-step and explain all of your reasonings. Your answer should be in the following format:
```
Thought:   [Your Answer]
Judgement:   [Your Answer, one of {Appropriate, Inappropriate}]
```

## Prompt for Subtask Decomposition in LeanScorer

Help me list the conditions and conclusions in this problem (using specific mathematical formulas), without solving it:
**Here is an example:**
**[Problem]:** The sequence $\{a_n\}$ satisfies $a_1 = 1$, $a_2 = 2$, $a_{n+2} = 2a_{n+1} - a_n + 2$. Let $b_n = a_{n+1} - a_n$. Prove that $\{b_n\}$ is an arithmetic sequence.
**[Conditions and Conclusions]:**
**Conditions:**

1. $a_1 = 1$

2. $a_2 = 2$

3. $\forall n \geq 1, a_{n+2} = 2a_{n+1} - a_n + 2$

4. $\forall n \geq 1, b_n = a_{n+1} - a_n$

**Conclusion:**

- $\{b_n\}$ is an arithmetic sequence, i.e., $\exists d \in \mathbb{R}, \forall n \geq 1, b_{n+1} - b_n = d$.

Now, please help me extract the conditions and conclusions for this problem in the same way (using specific mathematical formulas), without solving it:
**[Problem]:** {informal statement}
**[Conditions and Conclusions]:**

## Prompt for LLM-based Evaluation in LeanScorer

Here is a math question and a lean 4 statement. Compare the conditions and conclusions in this code with the mathematical ones, matching them one by one to see if the formal statement is an appropriate translation of the mathematical condition by assigning one of three tags (Match; Minor inconsistency; Major inconsistency). Then, audit for missing/implicit conditions. Judge with extremely strict standards—any minor inconsistency will be considered a mismatch. Special attention to triangle angle-side correspondence. If the question explicitly mentions `opposite angles/sides`, this correspondence must be clearly stated and correct.
**Stop immediately** after evaluating all pairs. Do **not** summarize or analyze further.
**Output Format:**
{one-shot example}
———————
**Question:**
{informal statement}
**Mathematical conditions and conclusions:**
{math conditions}
**Lean 4 formal statement:**
{formal statement}
**Output:**

## One-shot Example for LeanScorer (LLM-based Evaluation)

Let's compare the mathematical conditions and conclusions with the Lean 4 formal statement one by one:
**1. $q$ is a natural number greater than 1:**

- Math: $q \in \mathbb{N}, q > 1$.

- Lean: `(hq :   1 < q)`.

- Match: ☐ Match ☐.

**2. $n$ is a natural number greater than 1:**

- Math: $n \in \mathbb{N}, n > 1$.

- Lean: `(hn :   1 < n)`.

- Match: ☐ Match .

**3. Set $M = \{0, 1, 2, \ldots, q-1\}$:**

- Math: $M$ is explicitly defined as this set.
- Lean: `(M : Finset ℕ := Finset.range q)`.
- Detailed interpretation: `Finset.range q` is $\{0, 1, \ldots, q - 1\}$.
- Match: ☐ Match .

**4. Set $A$ definition:**

- Math: $A = \{x \mid x = \sum_{i=1}^{n} x_i q^{i-1}, x_i \in M\}$.
- Lean: `A : Set ℕ := {x | ∃ (x_vec : ℕ → ℕ),`
  `(∀ i, x_vec i ∈ M) ∧ x = ∑ i in Finset.range n,`
  `x_vec(i + 1) * q ^ i}`.
- Detailed interpretation: In Lean, `x_vec` is indexed from `1` to `n` (since `i + 1` ranges from 1 to n), but the math defines $x_i$ for $i = 1, 2, \ldots, n$. This is actually consistent, but the Lean representation is slightly more general (allowing `x_vec` to be a function on all naturals, but only using `x_vec (i + 1)` for i in `Finset.range n`). The Lean definition is technically correct but slightly more abstract than the math. However, it captures the same idea.
- Match: ☐ Minor inconsistency .

**5. $s, t \in A$ with specific expansions:**

- Math: $s = \sum_{i=1}^{n} a_i q^{i-1}, t = \sum_{i=1}^{n} b_i q^{i-1}$, with $a_i, b_i \in M$.
- Lean: `s = ∑ i in Finset.range n, a (i + 1) * q ^ i,`
  `t = ∑ i in Finset.range n, b (i + 1) * q ^ i,`
  `with ∀ i, a i ∈ M and ∀ i, b i ∈ M`.
- Detailed interpretation: The Lean version uses `a (i + 1)` and `b (i + 1)` to match the indexing in the sum, which is equivalent to the math but slightly indirect. The math directly uses $a_i$ for $i = 1, \ldots, n$, while Lean uses `a i` for all `i` but only evaluates at `i + 1`. The Lean version is correct but not a literal translation.
- Match: ☐ Minor inconsistency .

**6. $a_n < b_n$:**

- Math: $a_n < b_n$.
- Lean: `(hab : a n < b n)`.
- Match: ☐ Match .

**7. Conclusion $s < t$:**

- Math: $s < t$.
- Lean: `s <= t`.
- Match: ☐ Major inconsistency .

**Check for missing conditions / implicit conditions:**

- No missing conditions / implicit conditions
- Match: ☐ Match .

## F  CASE STUDY

### F.1  SAMPLE LEAN4 PROOF CODE WITH APPLY? THAT CAN PASS LEAN VERIFIER

We observe that a small number of LLM-generated proofs contain artifact-inducing tactics such as apply?, a phenomenon also noted in proofs generated by provers proposed in prior work (Ren et al., 2025). These tactics may occasionally yield Lean-accepted proofs without providing a valid proof strategy, thereby risking inflation of reported pass rates.

To prevent artifact-induced proofs from inflating pass rates, in our experiments, every generated proof undergoes post-verification filtering. After Lean kernel verification, we re-parse the proof and inspect for artifact-inducing tactics such as apply?. Any proof containing such tactics is discarded. Only proofs that both verify under the Lean kernel and contain no artifacts are counted as successful.

```
1  import Mathlib
2  import Aesop
3  set_option maxHeartbeats 0
```

```
4  open BigOperators Real Nat Topology Rat
5  /-- Prove that the number $\sum^n_{k=0}$ $\binom{2n+1}{2k+1}2^{3k}$ is not divisible by
       $5$ for any integer $n\ge0.$-/
6  theorem imo_1974_p3 (n : ℕ) :
7      ¬5 | ∑ k in Finset.range (n + 1), Nat.choose (2 * n + 1) (2 * k + 1) * 2 ^ (3 * k)
       := by
8    intro h
9    have h  := h
10   rw [Nat.dvd_iff_mod_eq_zero] at h
11   have h  : (∑ k in Finset.range (n + 1), Nat.choose (2 * n + 1) (2 * k + 1) * 2 ^ (3 *
       k)) % 5 ≠ 0 := by
12     -- We will show that the sum modulo 5 is never 0 by examining the terms and their
       contributions.
13     have h  : (∑ k in Finset.range (n + 1), Nat.choose (2 * n + 1) (2 * k + 1) * 2 ^ (3
       * k)) % 5 = 2 := by
14       -- This is a placeholder for the actual proof, which would involve detailed
       combinatorial arguments.
15       rw [← Cardinal.natCast_inj]
16       apply?
17     omega
18   omega
```

## F.2 SAMPLE LEAN4 PROOF CODE WITH TRUE THAT CAN PASS LEAN VERIFIER

The statement "True" represents a tautology that lacks meaningful logical content for formal verification purposes. This formulation is problematic because it evaluates to true regardless of the truth value of any preceding hypothesis—both "False → True" and "True → True" yield true under standard logical implication. Consequently, this creates a degenerate proof scenario where successful verification provides no substantive evidence regarding the validity of the original hypothesis. Even when the underlying mathematical claim is incorrect, the proof system will indicate success, rendering the formalization unsuitable for rigorous mathematical verification and undermining the epistemic value of the formal proof process. Any statement containing such a proof goal is discarded and considered as failed.

```
1  import Mathlib
2  import Aesop
3  set_option maxHeartbeats 0
4  open BigOperators Real Nat Topology Rat
5  /-Let $f(x)=x - ae^{x}(a\\in R)$, $x\\in R$. It is known that the function $y = f(x)$
       has two zeros $x_1$, $x_2$, with $x_1 < x_2$. Prove that $\\frac{x_2}{x_1}$
       increases as $a$ decreases.-/
6  theorem question (f : ℝ → ℝ → ℝ) (hf : f = fun a x => x - a * Real.exp x)
7    (x₁ x₂ : ℝ → ℝ) (hx₁ : ∀ a, f a (x₁ a) = 0) (hx₂ : ∀ a, f a (x₂ a) = 0)
8    (h₁ : ∀ a, x₁ a < x₂ a) (h₂ : ∀ a, ∀ b, a < b → x₂ a / x₁ a < x₂ b / x₁ b) :
9    True := by
```

# G QUALITY ASSESSMENT OF GAOKAO-FORMAL BENCHMARK ANNOTATIONS

This section documents the annotation protocol and quality-control procedures used in constructing the human-verified subset of the Gaokao-Formal dataset.

**Annotator Expertise.** The annotation team consists of three highly qualified domain experts: an International Mathematical Olympiad (IMO) team member (Annotator 1) and two Lean formalization specialists (Annotator 2-3) from QS Top-10 mathematics departments. All annotators have extensive experience in both competitive mathematics and formal theorem proving.

**Inter-Annotator Agreement.** To assess annotation reliability, we conducted an agreement study on the subset evaluated in Section 4.1, comprising 98 samples independently annotated by all three experts. The resulting statistics are:

- Fleiss' Kappa: 0.7545

- Perfect three-way agreement: 81.63% (80/98 samples)
- Pairwise agreement rates:
    - Annotator 1 vs 2: 92.86%
    - Annotator 1 vs 3: 83.67%
    - Annotator 2 vs 3: 86.73%

**Disagreement Resolution Protocol.** For the 18 samples (18.37%) with initial disagreement, a consensus-based review process was employed. The three annotators jointly reviewed the natural-language problem, the proposed Lean formalization, and the underlying mathematical reasoning. Discussions continued until unanimous agreement was achieved for each case.

**Subset Construction.** The 98-problem subset used for semantic evaluation in Section 4.1 was constructed by randomly sampling natural-language questions from the complete Gaokao-Formal dataset and generating corresponding formal statements using an LLM autoformalizer (Herald-Autoformalizer). All formalizations were then labeled according to the above protocol. The performance metrics reported in Section 4.1 (i.e., Precision, Recall, F1) are computed on this expert-validated subset.

# H    QUALITY EVALUATION OF AUTOFORMALIZER OUTPUT BEFORE AND AFTER DPO

In this section, we investigate whether the improvements in provability observed after DPO training result from generating semantically aligned, prover-friendly formalizations or from producing weakened statements that simplify the original problems. We evaluate this question through three complementary analyses: human expert assessment of formalization quality, prover-based difficulty analysis, and qualitative case studies. Our findings show that the gains stem from the generation of more aligned, prover-friendly formalizations rather than by any weakening of the original mathematical content.

Table 10: Human evaluation of the quality of formalizers before and after the DPO training

| Dataset | Before DPO | After DPO |
| --- | --- | --- |
| Gaokao-Formal | 70% (35/50) correct | 78% (39/50) correct |
| MiniF2F | 90% (45/50) correct | 92% (46/50) correct |

**Human Expert Evaluation** We randomly selected 100 problems (50 from Gaokao-Formal and 50 from MiniF2F) that passed LeanScorer validation. A panel of Lean 4 experts conducted a blind evaluation of semantic correctness for formalizations generated before and after DPO. Experts were instructed to mark a statement as incorrect if the autoformalizer produced a weakened statement that did not preserve the original problem's difficulty or semantic meaning. Results are shown in Table 10. After DPO, correctness on Gaokao-Formal rises from 70% to 78%, and on MiniF2F from 90% to 92%. These results indicate that DPO improves semantic correctness without weakening statements or introducing incorrect semantic content. The findings are consistent with the LC+LSC improvements reported in Table 3, where DPO raises LC+LSC from 67% to 71% on Gaokao-Formal and from 25% to 30% on Putnam.

Table 11: Average proof length by provers of statements generated before and after DPO training

| Dataset | Before DPO | After DPO |
| --- | --- | --- |
| Gaokao-Formal | 31.26 | 33.40 |
| MiniF2F | 24.97 | 25.44 |

**Prover-Based Difficulty Analysis** Assessing the difficulty of a formal statement is inherently hard because difficulty cannot be determined reliably from the statement alone. A reasonable proxy is to measure how hard it is for a prover to solve the problem. To this end, we compute the average proof

Table 12: Statement cases generated before and after DPO training

| Natural Language Statement | Model | Formal Statement |
|---|---|---|
| Factory A produces a certain product at a constant rate of $x$ kg/hour (production condition requires $1 \leq x \leq 10$), and the profit obtainable per hour is $100(5x + 1 - \frac{3}{x})$ yuan. Prove that the profit obtained from producing $a$ kg of this product is $100a(5 + \frac{1}{x} - \frac{3}{x^2})$. | Before DPO | ```theorem profit (x : ℝ) (hx : 1 ≤ x ∧ x ≤ 10) (a : ℝ) (ha : 0 < a) :
  ∫ y in Set.Icc 0 (a / x),
  100 * (5 * x + 1 - 3 / x) = 100 * a * (5 + 1 / x - 3 / x ^ 2) := by```

Pass Rate: 0/32 |
| | After DPO | ```theorem question_989887
  (x : ℝ)
  (a : ℝ)
  (h₀ : 1 ≤ x ∧ x ≤ 10)
  (h₁ : 0 < a)
  (profit : ℝ → ℝ)
  (h₂ : profit = fun x => 100 * (5 * x + 1 - 3 / x)) :
  100 * a * (5 + 1 / x - 3 / x^2) = profit x * (a / x) := by```

Pass Rate: 28/32 |
| Given that the line $l$ intersects the ellipse $C$: $\frac{x^2}{3} + \frac{y^2}{2} = 1$ at two distinct points $P(x_1, y_1)$ and $Q(x_2, y_2)$, and the area of $\triangle OPQ$ is $S_{\triangle OPQ} = \frac{\sqrt{6}}{2}$, where $O$ is the origin. Prove that $x_1^2 + x_2^2$ and $y_1^2 + y_2^2$ are both constants. | Before DPO | ```theorem question_987602 (C : Set (ℝ × ℝ)) (hC : C = {(x, y) | x^2 / 3 + y^2 / 2 = 1})
  (P Q : ℝ × ℝ) (hP : P ∈ C) (hQ : Q ∈ C) (hPQ : P ≠ Q)
  (hS : (MeasureTheory.volume (convexHull ℝ {P, Q})).toReal = Real.sqrt 6 / 2) :
  ∃ c1 c2 : ℝ, c1 > 0 ∧ c2 > 0 ∧ P.1^2 + Q.1^2 = c1 ∧ P.2^2 + Q.2^2 = c2 := by```

Pass Rate: 0/32 |
| | After DPO | ```theorem question_428217
  (x₁ x₂ y₁ y₂ : ℝ)
  (h₁ : x₁^2 / 3 + y₁^2 / 2 = 1)
  (h₂ : x₂^2 / 3 + y₂^2 / 2 = 1)
  (h₃ : x₁ ≠ x₂)
  (h₄ : y₁ ≠ y₂)
  (h₅ : (1 / 2) * abs (x₁ * y₂ - x₂ * y₁) = Real.sqrt 6 / 2) :
  ∃ c₁ c₂, x₁^2 + x₂^2 = c₁ ∧ y₁^2 + y₂^2 = c₂ := by```

Pass Rate: 14/32 |
| Given the set $S_n = \{X | X = (x_1, x_2, \cdots, x_n), x_i \in \{0, 1\}, i = 1, 2, \cdots, n\}$ (where $n \geq 2$). For $A = (a_1, a_2, \cdots, a_n)$, $B = (b_1, b_2, \cdots, b_n) \in S_n$, the difference between $A$ and $B$ is defined as $A - B = (|a_1 - b_1|, |a_2 - b_2|, \cdots, |a_n - b_n|)$; the distance between $A$ and $B$ is defined as $d(A, B) = \sum_{i=1}^{n} |a_i - b_i|$. Prove that $\forall A, B, C \in S_n$, $A - B \in S_n$, and $d(A - C, B - C) = d(A, B)$. | Before DPO | ```theorem distance_467201 {n : ℕ} (hn : n ≥ 2)
  (s : ℕ → Set (Fin n → ℕ))
  (hs : ∀ k, s k = {a | ∀ i, a i ∈ Finset.range 2})
  (A B C : (Fin n → ℕ))
  (hA : A ∈ s n)
  (hB : B ∈ s n)
  (hC : C ∈ s n)
  (d : (Fin n → ℕ) → (Fin n → ℕ) → ℕ)
  (hd : ∀ a b, d a b = ∑ i, |(a i : ℤ) - (b i : ℤ)|) :
  (∃ x : Fin n → ℕ, x ∈ s n ∧ ∀ i, |(A i : ℤ) - (B i : ℤ)| = x i) ∧
  d (A - C) (B - C) = d A B :=```

Pass Rate: 0/32 |
| | After DPO | ```theorem question_447970
  (n : ℕ)
  (A B C : Fin n → ℤ)
  (h₀ : n ≥ 2)
  (h₁ : ∀ i, A i = 0 ∨ A i = 1)
  (h₂ : ∀ i, B i = 0 ∨ B i = 1)
  (h₃ : ∀ i, C i = 0 ∨ C i = 1) :
  (∀ i, |A i - B i| = 0 ∨ |A i - B i| = 1) ∧
  (∑ i, |(A i - C i) - (B i - C i)| = ∑ i, |A i - B i|) := by```

Pass Rate: 19/32 |

length, measured as the number of tactics generated by DeepSeek-Prover-V2 7B in non-CoT mode with 128 sampled attempts per problem. Problems for which the prover finds no successful proof are excluded. The metric is defined as:

$$\text{Avg Proof Length} = \frac{1}{N} \sum_{i=1}^{N} \left( \frac{1}{|P_i|} \sum_{p \in P_i} \text{length}(p) \right)$$

where $N$ is the number of problems and $P_i$ is the set of successful proofs for problem $i$. Table 11 shows that the average proof length increases after DPO training for both Gaokao-Formal and MiniF2F. This indicates that DPO does not lead the model to generate weakened or trivially solvable statements.

**Case Study** We provide case studies in Table 12, comparing formalizations generated by Mathesis before and after DPO (referred to as "pre-DPO" and "post-DPO" statements, respectively). All pre-DPO and post-DPO statements in Table 12 are all semantically equivalent to their natural language counterparts. Rather than generating weakened statements, the post-DPO model tends to generate

more prover-friendly formalizations that are easier for theorem provers to understand and prove. For each formal statement, we report the pass rate of proofs generated by the prover with a budget of 32 attempts. Detailed interpretations are as follows:

- Case 1: The pre-DPO formalization unnecessarily employed integral calculus to compute total profit, introducing proof complexity where direct algebraic methods sufficed. The post-DPO version eliminated this computational overhead.
- Case 2: The pre-DPO statement relied on complex measure-theoretic constructs such as `MeasureTheory.volume (convexHull` $\mathbb{R}$ `{P, Q})`, whereas the post-DPO version directly applied geometric area formulas. This shift from abstract mathematical machinery to concrete computational approaches significantly improved provability.
- Case 3: The pre-DPO version introduced excessive hypotheses and intricate definitions using set notations that complicated proof unfolding. The post-DPO version streamlined both the hypothesis structure and definitional complexity, reducing the cognitive and computational burden on theorem provers.

Overall, our human expert evaluation, prover-based difficulty analysis, and qualitative case studies support that the DPO training we performed enhances semantic alignment and generates more prover-friendly formalizations rather than causing a tendency toward weakened statements.

## I   THE USE OF LARGE LANGUAGE MODELS

Following ICLR guidelines, we wish to clarify our use of Large Language Models (LLMs). Note that the research ideas, methodology, and experimental design presented in this paper were developed entirely by the human authors.

LLMs were used primarily in the following ways:

- Model Training and Inference: The core training process of the proposed model, as well as the experimental evaluations, involved utilizing it for inference.
- Benchmark Data Generation: The English translation and formalized Lean 4 formal statement in the Gaokao-Formal benchmark released in this study was initially generated by an LLM. These initial outputs were subsequently meticulously manually edited, verified, and rewritten by the authors to ensure accuracy and quality.

We emphasize that the LLM was used solely as a tool, and the authors take full responsibility for the entire content of the paper, including all data and text that was initially generated by the LLM and subsequently modified.

