# OpenReview forum: "Mathesis: Towards Formal Theorem Proving from Natural Languages"
_ICLR.cc/2026/Conference — ICLR 2026 Poster_

### Official Review · Reviewer_NXS9 · 2025-10-15

**Soundness:** 2
**Presentation:** 2
**Contribution:** 3
**Rating:** 4
**Confidence:** 3

**Summary:**

The authors propose Mathesis, the first autoformalizer trained with online reinforcement learning. Mathesis tackles the problem setting of formal theorem proving directly from informal problem statements. It is trained with GRPO to maximize semantic and syntactic correctness, and DPO to increase provability of the resulting formalization. The authors also introduce the Gaokao-Formal dataset of problems and LeanScorer to evaluate autoformalization correctness.

**Strengths:**

The paper uses online RL on the task of statement autoformalization for the first time, showing substantial gains upon the base model. It shows that GRPO with a syntactic check and LLM judge as reward can improve autoformalization.

Moreover, it demonstrates with DPO training that training the model to generate statements that are provable by a downstream prover can improve the quality of autoformalization.

The authors also propose LeanScorer, a more complicated LLM judger than a binary label, which they demonstrate gives higher precision.

I am also convinced that generating statements that are more easily proven is useful in the setting of end-to-end theorem proving (starting from an informal problem and ending with a formal proof), which is an interesting under-explored problem formulation that has many practical uses and overcomes the scarcity of expert-written formal statements.

**Weaknesses:**

I am not convinced by the complicated formulation of the Sugeno integral. The authors have not provided motivation to use the Sugeno integral, nor any explanation for the formula of $\mu$ (line 295). For example, why not just use this score: $0$ if there is a label M0, and otherwise use $\frac{n_{M1}}{|s|}$ as the score? Why choose the specific $\delta$ value and $\mu$ formula; is it to obtain a certain reward shape?

Also, it seems to me that the score $S$ is actually very simple:

- If there is a label M0 in $L$, then $f(l_{\pi(1)})=0$ so $\mu(s_i)=0$ for all $i$, so the score is $0$.
- Otherwise, sort the labels to $n_1$ labels of M½ and $n-n_1$ labels of M1. Then for $1 \le i \le n_1$, $\mu(s_i)=0$ because $n_{M1}=0$. For $i>n_1$, for the set $s_i$, $n_{M1/2}$ is fixed and $\frac{n_{M1}}{|s_i|} = \frac{i-n_1}{i}$ increases as $i$ increases, while $f(l_{\pi(i)}) = 1$, so the maximum on line 295 is always taken at $i=n$. Therefore the formula on line 295 simplifies to $S(L,f,\mu) = \frac{n-n_1}{n}(1-\delta n_1)$.

I concede that I am not familiar with Sugeno integral, and I could have misunderstood some details. I would be happy to dismiss this issue if the authors could explain the motivation for using Sugeno integral and their formulation, and point out my error. Otherwise I would suggest to give the formula for the score in simplified form, and to only give the fuzzy integral formulation if there is a well-founded motivation behind doing so.

In any case, LeanScorer seems like LLM-as-a-judge with a more complex prompt and score shaping. I am not convinced that LeanScorer is different from LLM-as-a-judge with a more carefully crafted prompt. For example, I think if we prompt the LLM to be stricter (e.g. by giving it a rubric) in Table 2, the precision of LLM-as-a-judge will increase and recall might decrease from 100%, and the F1 score might match LeanScorer, which is itself a LLM judge anyway. Also, LeanScorer is only used for miniF2F-test and Gaokao-Formal, but in both cases the human-written ground-truth formalization exists. Why doesn't the LLM simply compare to the ground-truth formalization, or use BEq (see my questions)?

I am not surprised that performance increased significantly on Gaokao-Formal in Table 3, because the RL training also used Gaokao problems, and the autoformalizer distribution could just be shifted toward Gaokao more.

It is unclear what “proving accuracy” (line 423) means to me. If I guess correctly, starting from an informal statement, proving accuracy is 1 if a prover model can prove the formal statement translated by another autoformalizer, and 0 otherwise. Is this correct? Consequently, the improvement in Figure 4 is hard to interpret. For example if the autoformalizer model gives wrong formalizations but they are easier to prove (with DPO training), this would still increase the “accuracy”.

Overall, some parts of the paper could be clearer and key ideas could be better conveyed. For example, the authors give many numerical comparisons in the text of Section 4.3, but little explanation, insight, or examples as to why the score of their model is higher. The only “case study” in Appendix E does not explain what it studies and is not referenced anywhere else.

I am happy to increase the score if the weaknesses and questions are addressed.

**Questions:**

Is there any overlap between the data used for training (“the natural language informal statements of problems from a pset (Lin et al., 2025) and our in-house Gaokao dataset”) and the miniF2F-test and Gaokao-Formal datasets used for evaluation? Since both training and testing use a subset of Gaokao problems, is there a chance of contamination?

What is Mathesis-Autoformalizer (without HPO) in Table 3? Is it the model after GRPO but before DPO training?

LeanScorer breaks down the natural language statement into subtasks, and compares against corresponding parts in the formal statement. However, what happens if there is an extra part in the formal statement that does not correspond to any natural language? For example, there could be a nondegeneracy criterion (a denominator is nonzero), or the model could have made a mistake and added an unneeded formal criterion. Do these unmatched parts exist and are they ignored?

The fact that the base model is Kimina-Autoformalizer (and that its size is probably 7B) should be mentioned in the main text instead of the appendix. For example, this makes it easier to understand Table 3.

There could be some comparison between LeanScorer and BEq/BEq+ (Liu et al. 2024 https://openreview.net/forum?id=hUb2At2DsQ) which has been researched in recent work in training and evaluating autoformalizers (Poiroux et al., 2024; Wu et al., 2025 https://arxiv.org/abs/2508.04440). Comparing against BEq which requires ground-truth labels also better contextualizes this work. I would like to see a comparison to BEq or BEq+ in Table 2. Since Gaokao-Formal contains ground-truth formal statements, I would also like to see BEq scores in Table 3, if BEq turns out to be a good scorer.

Minor suggestions:

- Table 2: the precision score of 94 instead of 93 should be bolded.
- Line 305: $e_{\pi(i)}$ -> $l_{\pi(i)}$.

---

> ### Author Response · Authors · 2025-11-22
> **Response to R5W2 and R5W3**
>
> **Response to R5W2**
>
> >_R5W2: In any case, LeanScorer seems like LLM-as-a-judge with a more complex prompt and score shaping. I am not convinced that LeanScorer is different from LLM-as-a-judge with a more carefully crafted prompt. For example, I think if we prompt the LLM to be stricter (e.g. by giving it a rubric) in Table 2, the precision of LLM-as-a-judge will increase and recall might decrease from 100\%, and the F1 score might match LeanScorer, which is itself a LLM judge anyway. Also, LeanScorer is only used for miniF2F-test and Gaokao-Formal, but in both cases the human-written ground-truth formalization exists. Why doesn't the LLM simply compare to the ground-truth formalization, or use BEq (see my questions)?_
>
> Thanks for the comment.  First, we note that equivalence-based checks such as BEq require access to a ground-truth formalization, which fundamentally limits scalability. In our task setting, given a real natural-language mathematical problem, Mathesis-Autoformalizer first generates a formal Lean statement, after which a prover attempts to produce a Lean-verified proof for that statement. Because no ground-truth formalization exists for most real-world problems, including Gaokao, olympiad, and arbitrary textbook questions, BEq cannot be applied. Obtaining ground-truth formalizations requires substantial human effort and mathematical expertise, which is precisely one of our core motivations for developing automated formalization and evaluation methods.
>
> Second, while both LeanScorer and LLM-as-a-Judge rely on LLMs, we define LLM-as-a-Judge as a single-prompt method that issues one holistic semantic-correctness decision. In contrast, LeanScorer follows a structured three-stage procedure: (1) LLM-assisted subtask decomposition, (2) LLM-assisted alignment and labeling of each subtask with its formal counterpart, and (3) principled aggregation of these labels into a final semantic-consistency score. This decomposition-based evaluation, rather than prompt complexity alone, is what enables LeanScorer to achieve a substantially better precision–recall trade-off.
>
> **Response to R5W3**
>
> >_R5W3: I am not surprised that performance increased significantly on Gaokao-Formal in Table 3, because the RL training also used Gaokao problems, and the autoformalizer distribution could just be shifted toward Gaokao more._
>
> Thanks for the comment. We agree that distinguishing between genuine reasoning capability and distribution shift is critical. To address this, we conducted a rigorous contamination analysis (detailed in the newly added Appendix B) using strict N-gram overlap detection. We found **zero** ``dirty'' matches (near-duplicates) between our training set and the Gaokao-Formal benchmark.
>
> It is important to clarify the distinction in data sources: our training data consists of _exercise problems_ collected from practice books and mock tests, whereas the Gaokao-Formal benchmark is constructed exclusively from _official questions_ administered in the National College Entrance Examination.
>
> Furthermore, our model's improvements are not limited to the Gaokao domain. As shown in Table 3, Mathesis-HPO also achieves state-of-the-art performance on MiniF2F-test (96\% LC+LSC@6) and PutnamBench (30\% LC+LSC@6, a 3$\times$ improvement over the baseline). This consistent cross-domain success demonstrates that our method enhances general formalization capabilities rather than merely shifting the distribution toward specific Gaokao patterns.

---

> ### Author Response · Authors · 2025-11-22
> **Response to R5W4**
>
> **Response to R5W4**
>
> >_R5W4: It is unclear what “proving accuracy” (line 423) means to me. If I guess correctly, starting from an informal statement, proving accuracy is 1 if a prover model can prove the formal statement translated by another autoformalizer, and 0 otherwise. Is this correct? Consequently, the improvement in Figure 4 is hard to interpret. For example if the autoformalizer model gives wrong formalizations but they are easier to prove (with DPO training), this would still increase the “accuracy”._
>
> Thanks for the comment.
>
> Yes. Proving accuracy is 1 if a prover model can prove the formal statement translated by another autoformalizer, and 0 otherwise.
>
> As shown in Figure 2, the Lean compilation (syntactic check) and LeanScorer validation (semantic check) process safeguards against misaligned statements that could otherwise introduce bias into downstream proving: all formalized statements must pass both syntactic and semantic checks before entering the proving stage. Table 3 reports the quality of formalized statements produced by different models, including both Lean Check and Lean Check combined with LeanScorer Semantic Check. Figure 4 then evaluates theorem proving from natural language by pairing each autoformalizer with multiple downstream provers and measuring proof success rates. Importantly, proof accuracy in Figure 4 is computed only on statements that have already passed both syntax and semantic validation.
>
> To directly address the concern that our model might generate easier statements after DPO training, we conducted additional experiments with 100 randomly sampled problems (50 from Gaokao-Formal, 50 from MiniF2F) that passed LeanScorer validation.
>
>
> **Human Expert Evaluation**
>
> We ran a blind evaluation with a Lean 4 expert who assessed the semantic correctness of formalizations before and after DPO. Results are shown in the table below:
>
> | Dataset | Before DPO | After DPO |
> |---------|------------|-----------|
> | Gaokao-Formal | 70\% (35/50) correct | 78\% (39/50) correct |
> | MiniF2F | 90\% (45/50) correct | 92\% (46/50) correct |
>
> We observe that DPO training improves semantic correctness rather than sacrificing it for easier statements. This is consistent with our LC+LSC improvements in Table 3 (Mathesis 67\% to Mathesis-HPO 71\% on Gaokao-Formal, and from 25\% to 30\% on Putnam).
>
> **Prover-Based Difficulty Analysis**
>
> We recognize that objectively defining ``difficulty" for semantically correct statements is fundamentally challenging—without solving the problem, it is difficult to assess difficulty from the statement alone. As a proxy, we measured the average proof length (number of tactics) generated by DeepSeek-Prover-V2 7B (non-CoT mode, 128 samples per problem, excluding problems with no successful proofs). The metric is computed as:
>
> $$\text{Avg Proof Length} = \frac{1}{N} \sum_{i=1}^{N} \left( \frac{1}{|P_i|} \sum_{p \in P_i} \text{length}(p) \right)$$
>
> where $N$ is the number of problems and $P_i$ is the set of successful proofs for problem $i$. The results are as follows.
>
> | Dataset | Before DPO | After DPO |
> |---------|------------|-----------|
> | Gaokao-Formal | 31.26 | 33.40 |
> | MiniF2F | 24.97 | 25.44 |
>
> According to the results, proof complexity does not decrease after DPO. This does not support the hypothesis that DPO biases toward easier statements.
>
> **Case Study**
>
> We further provide one illustrative example (MiniF2F problem mathd_numbertheory_222), comparing the formalizations before and after DPO:
>
> Before DPO:
> ```lean
> theorem number_theory_969 :
>   {x : ℕ | x ≠ 120 ∧ Nat.lcm x 120 = 3720 ∧ Nat.gcd x 120 = 8} = {248}
> ```
>
> After DPO:
> ```lean
> theorem number_theory_447284 (x y : ℕ)
>   (h₀ : Nat.lcm x y = 3720) (h₁ : Nat.gcd x y = 8) (h₂ : x = 120) :
>   y = 248
> ```
>
> The above two statements are semantically equivalent to the natural-language statement, and the second formalization is easier to prove (integer equality vs. set equality). This reflects improved formalization, where the model produces a clearer and more canonical formulation that better matches Lean’s proof structure, which is precisely the goal of autoformalization, not a flaw.
>
> Overall, the results from our human expert evaluation, prover-based difficulty analysis, and qualitative examples support the conclusion that DPO improves semantic alignment rather than causing a tendency to generate weakened statements. We will release all evaluation data, including expert annotations and proof logs, for transparency and community verification.

---

> ### Author Response · Authors · 2025-11-22
> **Response to R5W5, R5Q1, R5Q2, R5Q3 and R5Q4**
>
> **Response to R5W5**
>
> >_R5W5: Overall, some parts of the paper could be clearer and key ideas could be better conveyed. For example, the authors give many numerical comparisons in the text of Section 4.3, but little explanation, insight, or examples as to why the score of their model is higher. The only “case study” in Appendix E does not explain what it studies and is not referenced anywhere else._
>
> Thanks for the suggestion. In the revised manuscript, we have strengthened Section 4.3 by adding clear explanations and insights that connect the numerical results, making the performance differences more interpretable. We have also expanded and clarified the case studies (now Appendix F) and added explicit explanations. These cases illustrate how certain provers may bypass the Lean verifier, an issue we identified during our analysis. We include these examples in the Appendix to benefit the community and to encourage more rigorous verification practices in future work.
>
> Please kindly refer to Section 4.3 and Appendix F; the added content is highlighted in blue.
>
> **Response to R5Q1**
>
> >_R5Q1: Is there any overlap between the data used for training (“the natural language informal statements of problems from a pset (Lin et al., 2025) and our in-house Gaokao dataset”) and the miniF2F-test and Gaokao-Formal datasets used for evaluation? Since both training and testing use a subset of Gaokao problems, is there a chance of contamination?_
>
> Thanks for the comment. We confirm that there is no overlap between the training and evaluation datasets. To verify this rigorously, we conducted a data contamination analysis (detailed in Appendix B) using strict N-gram overlap detection. The analysis revealed **zero** ``dirty'' matches (near-duplicates) between our in-house Gaokao training set and the Gaokao-Formal benchmark.
>
> The lack of contamination stems from the distinct sources of the data:
>
>  **Training Data:** Our in-house Gaokao dataset consists of _exercise problems_ sourced from practice books and mock examinations.
>
>  **Evaluation Data:** The Gaokao-Formal benchmark is constructed exclusively from _official problems_ administered in the actual National College Entrance Examination (Gaokao).
>
> These two sources are entirely distinct. Furthermore, as noted in the response to R4W1, our model also achieves state-of-the-art performance on MiniF2F and significantly improves transfer performance on PutnamBench, demonstrating robust generalization capabilities beyond the specific distribution of the Gaokao dataset.
>
> **Response to R5Q2**
>
> >_R5Q2: What is Mathesis-Autoformalizer (without HPO) in Table 3? Is it the model after GRPO but before DPO training?_
>
> Thank you for the question. Yes, Mathesis-Autoformalizer (without HPO) refers to the model obtained after the GRPO stage and before DPO training, whereas Mathesis-Autoformalizer-HPO corresponds to the model obtained after the full two-stage Hierarchical Preference Optimization (GRPO followed by DPO). To make this clearer, we have clarified it in Section 4.2 of the revised manuscript, with the added content highlighted in blue.
>
> **Response to R5Q3**
>
> >_R5Q3: LeanScorer breaks down the natural language statement into subtasks, and compares against corresponding parts in the formal statement. However, what happens if there is an extra part in the formal statement that does not correspond to any natural language? For example, there could be a nondegeneracy criterion (a denominator is nonzero), or the model could have made a mistake and added an unneeded formal criterion. Do these unmatched parts exist and are they ignored?_
>
> Thanks for the question. In the first stage, LeanScorer performs LLM-assisted subtask decomposition to decompose the full informal statements into a set of semantic units. Second, it aligns each subtask with its counterpart in the formal statement and assigns a label, assisted by LLM. Any additional assumptions, side conditions, or extra formal content in the formalization that do not correspond to any NL subtask are explicitly treated as misaligned; conversely, any NL subtask that is missing from the formalization is also marked as misaligned. We have clarified this in Section 3.2 of our revised paper.
>
> **Response to R5Q4**
>
> >_R5Q4: The fact that the base model is Kimina-Autoformalizer (and that its size is probably 7B) should be mentioned in the main text instead of the appendix. For example, this makes it easier to understand Table 3._
>
> Thank you for the suggestion. We have added this detail into the main text of the revised manuscript: In Section 3.1, we now explicitly state that our model is initialized from Kimina-Autoformalizer, a 7B SFT autoformalizer that served as the strongest publicly available baseline prior to our work. To make it easier to understand Table 3, we also note in Section 4.2 that Kimina-Autoformalizer is used as the base model for both Mathesis variants. The revised content is highlighted in blue.

---

> ### Author Response · Authors · 2025-11-22
> **Response to R5Q5 and R5Q6**
>
> **Response to R5Q5**
>
> >_R5Q5: There could be some comparison between LeanScorer and BEq/BEq+ (Liu et al. 2024 https://openreview.net/forum?id=hUb2At2DsQ) which has been researched in recent work in training and evaluating autoformalizers (Poiroux et al., 2024; Wu et al., 2025 https://arxiv.org/abs/2508.04440). Comparing against BEq which requires ground-truth labels also better contextualizes this work. I would like to see a comparison to BEq or BEq+ in Table 2. Since Gaokao-Formal contains ground-truth formal statements, I would also like to see BEq scores in Table 3, if BEq turns out to be a good scorer._
>
> We thank the reviewer for the suggestion.
>
> **Current Status**: We have begun efforts to reproduce BEq, but we currently encounter data-format incompatibility issues. In particular, BEq requires structured fields beyond the formal statement (e.g., hard-dependencies, mathlib-dependencies, proof-state, and header), and the authors have not released the corresponding necessary preprocessing scripts. As a result, executing their method is not yet feasible. We are in active communication with the authors and expect to resolve these issues soon.
>
> **Methodological Differences**: We note that BEq and LeanScorer have distinct use cases:
> - BEq is a supervised method that requires the ground-truth formal statements and a substantial set of auxiliary fields. These requirements limit scalability in realistic autoformalization scenarios, where ground-truth formalizations and such structured metadata are typically unavailable.
> - LeanScorer, in contrast, is unsupervised and ground-truth–free, making it directly applicable to our task setting, which aims to generate formal proofs from natural-language mathematical problems in real-world scenarios where no ground-truth formalization is available.
> - Both approaches have distinct use cases, and we will make this explicit in the revised paper.
>
> **Planned Actions**:
> 1. We will cite the paper and incorporate a discussion that BEq's supervised, ground-truth–dependent approach with LeanScorer's unsupervised, ground-truth–free approach.
> 2. Once we receive the necessary code from the authors and resolve preprocessing compatibility, we will add BEq results in our paper.
>
> We appreciate the reviewer's patience as we work to include this important comparison.
>
>
> **Response to R5Q6**
>
> >_R5Q6: Minor suggestions_
>
> We thank the reviewer for pointing out these typos. We have corrected the mistakes accordingly.

---

> ### Author Response · Authors · 2025-11-24
> **Response to R5W1 - Part 1**
>
> **Response to R5W1**
>
> >_R5W1: I am not convinced by the complicated formulation of the Sugeno integral. The authors have not provided motivation to use the Sugeno integral, nor any explanation for the formula of  (line 295). For example, why not just use this score: if there is a label M0, and otherwise use $\frac{n_{M1}}{|s|}$ as the score? Why choose the specific value and formula; is it to obtain a certain reward shape?
> I concede that I am not familiar with Sugeno integral, and I could have misunderstood some details. I would be happy to dismiss this issue if the authors could explain the motivation for using Sugeno integral and their formulation, and point out my error. Otherwise I would suggest to give the formula for the score in simplified form, and to only give the fuzzy integral formulation if there is a well-founded motivation behind doing so._
>
> Thank you for the comment and your careful review of the formula. We apologize for the confusion caused by the mistakes in the original mathematical expression, which may mislead readers into thinking that this score could be simplified. The implementation used in our experiments is correct; the errors arose solely during transcription into the manuscript. These issues have been corrected in the revised paper and are highlighted in blue in Section 3.2. We have also expanded the explanation of the motivation for using the Sugeno integral to clarify its necessity and rationale. Please find our response below.
>
> 1. Correction of the formula
>
> First, for the definition of $\mu(s)$ in Equation (1), we corrected the term $\dfrac{n_{\text{M1}}}{n}$ to $\dfrac{n_{s}}{n}$, where $n_s$ denotes the size of the evaluated subtask set $s$ and $n$ is the total number of subtasks labelled.  The revised definition of $\mu$ is:
>
> - $ \mu(s) = 0,$ if $\exists l \in s, l = M0, $
>
> - $\mu(s) =$ $\max$ {$\frac{n_{s}}{n} \cdot (1 - \delta \cdot n_{M½} )$, 0 } , otherwise
>
>
> Second, we corrected the definition of the set $s_i$ to be the suffix set instead of the prefix set. The revised definition now states: "define the suffix set $ s_i = \\{ l_{\pi(i)}, \dots, l_{\pi(n)} \\}$ , which represents the subset consisting of $l_{\pi(i)}$ and all labels ranked after it".
>
> To further illustrate how the Sugeno integral operates, we would like to provide an example: Let $L = \\{\text{M1}, \text{M1}, \text{M1}, \text{M½}, \text{M½}\\}$ be the set of subtask labels, with size $n=5$ and indices $[0,1,2,3,4]$.
>
>
> Step 1: Sort the labels in ascending order of labels ($\text{M½} < \text{M1}$), yielding the sorted sequence $\\{\text{M½}, \text{M½}, \text{M1}, \text{M1}, \text{M1}\\}$ with corresponding permutation $\pi = [3, 4, 0, 1, 2]$.
>
> Step 2: Compute Sugeno integral via the max-min formula:
> For each $i$ from $0$ to $1$, we take the label $l_{\pi(i)}$ whose original index is $\pi(i)$ before sorting and the suffix set $s_i = \\{ l_{\pi(i)}, \dots, l_{\pi(n)} \\}$. We then compute $\mu(s_i)$ and then evaluate $\min\left( f(l_{\pi(i)}),\ \mu(s_i) \right)$. The table below illustrates the full computation:
>
>
> | i | index of $s_i$ | Labels in $s_i$ | $\mu(s_i)$ | $f(l_{\pi(i)})$ | min($\mu$, f) |
> |---|-------------------|--------|---------|---------|-----------|
> | 0 | {3,4,0,1,2}      | {$\text{M½}$,$\text{M½}$,$\text{M1}$,$\text{M1}$,$\text{M1}$} | 5/5×(1-0.2×2)=0.6 | f($\text{M½}$)=0.5 | 0.5 |
> | 1 | {4,0,1,2}        | {$\text{M½}$,$\text{M1}$,$\text{M1}$,$\text{M1}$}   | 4/5×(1-0.2×1)=0.72| f($\text{M½}$)=0.5 | 0.5 |
> | 2 | {0,1,2}          | {$\text{M1}$,$\text{M1}$,$\text{M1}$}     | 3/5×1=0.6        | f($\text{M1}$)=1.0 | 0.6 |
> | 3 | {1,2}            | {$\text{M1}$,$\text{M1}$}       | 2/5=0.4          | f($\text{M1}$)=1.0 | 0.4 |
> | 4 | {2}              | {$\text{M1}$}         | 1/5=0.2          | f($\text{M1}$)=1.0 | 0.2 |
>
> Finally, the Sugeno integral is the maximum over all these values: $S = \max\\{0.5, 0.5, 0.6, 0.4, 0.2\\} = 0.6$.

---

> ### Author Response · Authors · 2025-11-24
> **Response to R5W1 - Part 2**
>
> **Response to R5W1 - Part 2**
>
> 2. Motivation for using Sugeno Integral
>
> We adopted the Sugeno integral as a principled multi-criteria decision-making  (MCDM) framework that: (1) aggregates ordinal subtask assessments into continuous  [0,1] scores, and (2) allows domain knowledge to be encoded through a customizable fuzzy measure.
>
> Our fuzzy measure is designed based on empirical label statistics:
>
> | Assessment | Incorrect | Correct | Ratio |
> |------------|-----------|---------|-------|
> | M1         | 48.7\%    | 88.3\%  | 1:1.81 |
> | M½         | 16.2\%    | 8.6\%   | 1.88:1 |
> | M0         | 35.2\%    | 3.1\%   | **11.4:1** |
> | All-M1     | 2.3\%     | 60.0\%  | **1:26.1** |
>
> These observations motivate the following design choices:
> - **Veto for M0**: M0 is overwhelmingly associated with incorrect formalizations (11× more frequent in incorrect cases), we let $\mu(s)=0$ whenever $s$ contains any M0 label.
>
> - **Penalty for multiple M½**: Since M½ appears almost 2× as often in incorrect cases, we introduce additional penalties when there are two or more M½ labels.
> - **Reward for all-M1**: All-M1 formalizations are 26× more likely to be correct, so scores approach 1.0 to reflect the strong correlation between all-M1 subtasks and correctness.
>
> Although our current fuzzy measure design is relatively simple, the Sugeno framework provides a theoretically sound and extensible framework that incorporates domain knowledge and allows future refinements. Empirical results (the next point) show competitive performance with superior robustness compared to simpler baselines.
>
> 3. Empirical Comparison with Simpler Alternatives
>
> We conducted ablation studies comparing the Sugeno integral with four baseline aggregation methods:
> - Binary score
> - Uniform averaging
> - Geometric mean, and
> - Ordered weighted averaging (OWA)
>
> The key findings (full results in Appendix D) are summarized below:
>
> | Methods | Best Threshold | Precision | Recall | F1 | Accuracy | Average Acc |
> |---------|----------------|-----------|--------|-----|----------|-------------|
> | Sugeno (ours) | 0.6 | 0.94 | 0.89 | 0.92 | 0.91 | 0.85(±0.046) |
> | Binary | N/A | 0.97 | 0.6 | 0.74 | 0.77 | 0.77 (±0.0) |
> | Uniform | 0.8 | 0.94 | 0.89 | 0.92 | 0.91 | 0.72(±0.117) |
> | Geometric | 0.6 | 0.89 | 0.89 | 0.89 | 0.87 | 0.82 (±0.069) |
> | OWA | 0.9 | 0.91 | 0.89 | 0.90 | 0.89 | 0.66 (±0.105) |
>
>
>
> The results show that the Sugeno integral offers:
> 1. Competitive or superior F1/accuracy compared to all baselines.
> 2. Significantly lower variance, indicating more stable performance across different test conditions.
> 3. Effective performance across a wider range of hyperparameter settings (performing reliably for $\delta \in [0.1, 0.7]$), whereas baselines are more sensitive to tuning.
>
> These results show that the added complexity of the Sugeno integral yields tangible practical benefits, particularly in robustness—an essential property for automated evaluation systems.
>
> We contend that the Sugeno integral offers a theoretically grounded and empirically validated framework for this task. The corrected formulation, combined with our ablation studies, demonstrates its advantages over simpler aggregation methods. We remain open to further suggestions for refining the fuzzy measure design and welcome additional insights from the community.

---

> > ### Comment · Reviewer_NXS9 · 2025-11-24
> >
> > Thank you for the detailed response. I think the authors have adequately responded to most of my concerns. In particular, the authors clarified their definition of LLM judge in their revised manuscript, and expanded on the motivation and corrected their formulation of the Sugeno integral. I encourage the authors to add the blind evaluation results (response to R5W4) to the appendix. I am satisfied with the expanded explanations throughout the paper during the revision. I am happy with the responses to my questions.
> >
> > I am still not convinced Sugeno integral is necessarily better than the baselines the authors used. For example, the very simple "uniform" baseline has exactly the same recall, precision, F1, and accuracy scores, and it is only lacking in "average accuracy". However, I don't think comparing sensitivity to a threshold value varying uniformly in [0.1, 1.0] is really fair, because different methods have intrinsically different output score distributions. But it is shown that the Sugeno integral formulation is at least as good as other methods and I am now satisfied with the decision of using it.
> >
> > Given the low Lean Check scores in the additional PutnamBench column in Table 3, showing the models having a low LC score, I am wondering if an iterative refinement (giving compiler feedback to the formalizer) would work better.
> >
> > Looking at the baseline API models in Table 3 for the Gaokao-Formal column, it seems that Mathesis-Autoformalizer is mainly better at making Lean statements pass LC, rather than making them pass LSC: the LC+LSC score divided by the LC score (which I interpret as the true LSC score, i.e. the semantically correct statements out of all syntactically correct ones) seems similar between API models and Mathesis-Autoformalizer. Perhaps with iterative refinement or higher pass@k, API models can get Lean statements to pass LC, and then the LSC score might surpass the trained models.
> >
> > It seems that Table 2 is still not revised (94 is not bolded).
> >
> > I have revised my score.

---

> > > ### Author Response · Authors · 2025-11-28
> > >
> > > We sincerely appreciate the reviewer's positive assessment of our work and for raising the ranking score. We value the constructive feedback provided, which has helped us clarify key aspects of our methodology and results.
> > >
> > > We appreciate the suggestion regarding test-time scaling strategies, such as increased sampling budgets and iterative refinement. To explore this, we conducted preliminary experiments with higher pass-k settings (k = 6, 16, 32) for both DeepSeek-V3—which achieves the strongest performance among all tested API models in our experiments—and Mathesis-Autoformalizer-HPO. The results are presented in the table below. The results illustrate that higher k primarily benefits both syntactic correctness and semantic correctness. Notably, the LC+LSC performance gap measured by LC+LCS between Mathesis and DeepSeek-V3 consistently increases as k increases, and DeepSeek-V3 does not surpass our Mathesis model at any k setting, further demonstrating the effectiveness of our model and training method. Regarding iterative refinement, such as leveraging compiler or checker feedback to guide subsequent formalization attempts, we consider this a promising direction for future work.
> > > | Model | k | LC | LC+LSC |
> > > |------|------|------|--------|
> > > | Mathesis-Autoformalizer-HPO | 6 | 73 | 30 |
> > > |   | 16 | 79 | 37 |
> > > |   | 32 | **85** | **44** |
> > > | Deepseek-V3 | 6 | 22 | 10 |
> > > |   | 16 | 27 | 13 |
> > > |   | 32 | 35 | 17 |

---

### Official Review · Reviewer_UPch · 2025-10-25

**Soundness:** 3
**Presentation:** 3
**Contribution:** 3
**Rating:** 6
**Confidence:** 4

**Summary:**

This paper proposes Mathesis, a pipeline for natural-language -> formal-statement -> Lean proof. The core components are: (i) Mathesis‑Autoformalizer, trained with online RL (GRPO) using a composite reward from Lean compilation (syntax) and an LLM-based semantic check, followed by a DPO stage aligned to downstream proof success (“Hierarchical Preference Optimization, HPO”); (ii) LeanScorer, an LLM-driven semantic evaluation with subtask decomposition and aggregation via the Sugeno fuzzy integral; and (iii) Gaokao‑Formal, a 495‑problem benchmark from Chinese college-entrance exams aimed at harder-to-formalize statements. On MiniF2F and Gaokao‑Formal, the autoformalizer improves pass rates in both autoformalization and theorem proving (paired with another prover model).

**Strengths:**

1. End-to-end focus: Unlike the purely formal models like DeepSeek-Prover and datasets like MiniF2F and FIMO, this paper attempts end-to-end full NL input problem -> formal proof pipeline and shows that better formalization substantially boosts downstream proving across multiple provers. Also, Gaokao‑Formal intentionally targets hard-to-formalize statements and broad topics (e.g., analytic geometry, comprehensive questions), which is valuable given miniF2F’s focus.

2. The composite syntax + semantics reward in GRPO followed by DPO aligned to proof success (HPO) is coherent; ablations show small but consistent gains. The autoformalizer attains higher LC and LC+LSC on both datasets than strong baselines like Kimina‑Autoformalizer, and improves pass@32 with multiple 7B prover models

**Weaknesses:**

1. Data deduplication / de-contamination: Sec. 3.1 states that training-data curation includes the in‑house Gaokao dataset plus Lean Workbook (yielding ~32k problems), and the main evaluation benchmark is also Gaokao‑Formal. Without explicit deduplication auditing (fuzzy NL/FL matching, near-duplicate detection, year‑wise splits), gains on Gaokao‑Formal may reflect domain adaptation or overlap. The paper should document dedup guarantees, overlap statistics, and controlled cross‑domain tests. Also, some of the problems are created before the knowledge cutoff data of the evaluated models, causing the data contamination issues. It would be nice to create a subset of uncontaminated examples for evaluation via temporal split.

2. While LeanScorer is thoughtful, the main LC+LSC@k results overly rely on proposed LLM‑based scorer, raising concerns about metric coupling. Recent alternatives specifically targeting semantic alignment include FormalAlign [1]. The paper compares LeanScorer only to LLM-as-a-judge and re‑informalization rather than to these stronger contemporaries, which this limits external validity.

[1] Lu, Jianqiao, et al. "Formalalign: Automated alignment evaluation for autoformalization." arXiv preprint arXiv:2410.10135 (2024).

**Questions:**

1. How does Mathesis‑Autoformalizer compare to CriticLean [1] and FormaRL [2] when both are run under your evaluation?

2. How does Gaokao‑Formal relate in difficulty and topic mix to miniF2F, PutnamBench, and ProofNet (e.g., cross‑benchmark transfer results or controlled difficulty calibration)?

3. What safeguards ensure the reported Lean pass rates are not inflated by `apply?` /True/circular-goal artifacts?

[1] Peng, Zhongyuan, et al. "Criticlean: Critic-guided reinforcement learning for mathematical formalization." arXiv preprint arXiv:2507.06181 (2025).

[2] Huang, Yanxing, et al. "Formarl: Enhancing autoformalization with no labeled data." arXiv preprint arXiv:2508.18914 (2025).

---

> ### Author Response · Authors · 2025-11-21
> **Response to R4W1**
>
> **Response to R4W1**
>
> >_R4W1: Data deduplication / de-contamination: Sec. 3.1 states that training-data curation includes the in‑house Gaokao dataset plus Lean Workbook (yielding ~32k problems), and the main evaluation benchmark is also Gaokao‑Formal. Without explicit deduplication auditing (fuzzy NL/FL matching, near-duplicate detection, year‑wise splits), gains on Gaokao‑Formal may reflect domain adaptation or overlap. The paper should document dedup guarantees, overlap statistics, and controlled cross‑domain tests. Also, some of the problems are created before the knowledge cutoff data of the evaluated models, causing the data contamination issues. It would be nice to create a subset of uncontaminated examples for evaluation via temporal split._
>
> Thanks for the comment. We agree that rigorous deduplication is essential to ensuring that the reported performance gains reflect genuine reasoning capabilities rather than mere memorization. To address your concern, we conducted a comprehensive contamination analysis using strict N-gram overlap detection. Below, we clarify the distinction between our data sources and present empirical evidence demonstrating that the Gaokao-Formal benchmark is not contaminated by our training sets. We have added the decontamination statistics and the detailed methodology to the Appendix of the revised paper.
>
> First, it is important to distinguish the sources of our data. Our training data, specifically the "in-house Gaokao dataset," consists of practice problems collected from exercise books and mock tests. These are distinct from the evaluation data in Gaokao-Formal, which is composed exclusively of official questions from the Chinese National College Entrance Examination (Gaokao) spanning 2008–2024.
>
> To rigorously verify this distinction and audit for potential overlap, we implemented a lexical overlap detection pipeline. The pipeline normalized all text (NFKC normalization, lowercase, whitespace collapse) and extracted all 50-character substrings starting at word boundaries from the evaluation benchmarks. We then utilized an Aho-Corasick automaton to stream the training corpora and detect matches. For each problem $i$, we calculated an overlap ratio $\eta_i$ defined as:
>
> $$\eta_i = \frac{\text{matched windows}}{\text{total windows}}$$
>
> Based on this metric, we categorized problems as Clean ($\eta < 0.2$), Suspicious ($0.2 \le \eta < 0.8$), or Dirty ($\eta \ge 0.8$).
>
> We evaluated Gaokao-Formal against our three primary training data sources: the Combined Corpus (English informal statements), the Goedel P-Set, and the Lean Workbook. As shown in the table below, Gaokao-Formal contains zero "Dirty" matches across all training sets. The "Suspicious" matches are negligible (ranging from 0.0\% to 3.2\%), and a manual inspection reveals that these overlaps typically involve standard mathematical phrasing (e.g., "Let $f(x)$ be a function defined on...") rather than specific problem leakage.
>
> | Evaluation Set | Training Source Checked | Total Entries | Clean (<0.2) | Suspicious ($0.2 \le \eta < 0.8$) | Dirty ($\ge 0.8$) |
> | :--- | :--- | :--- | :--- | :--- | :--- |
> | Gaokao-Formal | Combined Corpus | 495 | 489 (98.8\%) | 6 (1.2\%) | 0 (0.0\%) |
> | Gaokao-Formal | Goedel P-Set | 495 | 479 (96.8\%) | 16 (3.2\%) | 0 (0.0\%) |
> | Gaokao-Formal | Lean Workbook | 495 | 495 (100\%) | 0 (0.0\%) | 0 (0.0\%) |
>
> To further validate the sensitivity of our detection method, we applied the same analysis to existing benchmarks against the Goedel P-Set. In that control test, the tool successfully identified significant contamination in MiniF2F (31 dirty proofs) and PutnamBench (41 dirty proofs). This contrast highlights that Gaokao-Formal provides a significantly cleaner signal for evaluating generalization than these existing benchmarks within the context of current large-scale training data.

---

> ### Author Response · Authors · 2025-11-21
> **Response to R4W2, R4Q1 and R4Q2**
>
> **Response to R4W2**
>
> >_R4W2: While LeanScorer is thoughtful, the main LC+LSC@k results overly rely on proposed LLM‑based scorer, raising concerns about metric coupling. Recent alternatives specifically targeting semantic alignment include FormalAlign [1]. The paper compares LeanScorer only to LLM-as-a-judge and re‑informalization rather than to these stronger contemporaries, which this limits external validity. [1] Lu, Jianqiao, et al. "Formalalign: Automated alignment evaluation for autoformalization." arXiv preprint arXiv:2410.10135 (2024)._
>
> Thanks for the comment. FormalAlign operates by fine-tuning a dedicated alignment model and computing an alignment score for each pair of informal input and formal output. The score combines (i) a certainty score derived from the model’s exponential of average log-likelihood of the formal output, and (ii) a similarity score measuring cosine similarity between the hidden states of the informal input and the formal output conditioned on the informal input. However, because FormalAlign’s fine-tuned model and evaluation pipeline are not publicly available, it is not currently feasible to include a direct empirical comparison on our benchmarks.
>
> We would like to mention that LeanScorer is designed as a training-free semantic checker that does not require fine-tuning an additional model. In Section 2, we compare LeanScorer with state-of-the-art LLM-based, training-free verification methods. We have cited the FormalAlign (Lu et al., 2024a) in the related work section and added a methodological comparison in blue in the revised paper.
>
>
> **Response to R4Q1**
>
> >_R4Q1: How does Mathesis‑Autoformalizer compare to CriticLean [1] and FormaRL [2] when both are run under your evaluation? [1] Peng, Zhongyuan, et al. "Criticlean: Critic-guided reinforcement learning for mathematical formalization." _
>
> Thank you for the question. We provide comparisons with these two concurrent works. CriticLean is a concurrent work that focuses on formal statement semantic checking. We evaluated their critic models on our LeanScorer testset to provide a direct comparison. The results are as follows:
>
> CriticLeanGPT-Qwen3-32B-RL: Precision 0.82, Recall 0.42, F1 0.55, Accuracy 0.62
>
> CriticLeanGPT-Qwen3-8B-RL: Precision 0.76, Recall 0.51, F1 0.61, Accuracy 0.63
>
> In comparison, our LeanScorer achieves: Precision 0.94, Recall 0.89, F1 0.92, Accuracy 0.91, demonstrating superior performance across all metrics.
>
> Another concurrent is FormalRL. It develops formalizer models but has not released its models publicly. Based on the results reported in their paper, their best model achieves 86.27\% SC (equivalent to our LC metric) and 59.63\% CC (equivalent to our LC+LSC metric) on miniF2F at pass@1. In comparison, our best model reaches 99\% LC and 96\% LC+LSC at pass@1.
> We note that there may be differences in semantic verification methodologies between the two papers. To ensure a fair comparison, we focus on the LC metric, which represents syntax checking by the Lean compiler. By this measure, our Mathesis-Autoformalizer demonstrates stronger performance.
>
> **Response to R4Q2**
>
> >_R4Q2: How does Gaokao‑Formal relate in difficulty and topic mix to miniF2F, PutnamBench, and ProofNet (e.g., cross‑benchmark transfer results or controlled difficulty calibration)?_
>
> Thanks for the comment. In terms of educational level and problem composition, Gaokao-Formal differs from MiniF2F, PutnamBench, and ProofNet. Gaokao-Formal is constructed directly from official Gaokao examination problems and therefore reflects a broad distribution of high-school mathematics domains, including functions, sequences and series, inequalities, trigonometry, analytic geometry, probability/combinatorics (see Table 1). It also contains multi-step “comprehensive questions’’ that integrate concepts across domains, mirroring the structure of the Gaokao exam. By contrast, MiniF2F consists of exercise statements from olympiads and high-school and undergraduate maths classes, PutnamBench is sourced from premier undergraduate-level mathematics competition, and ProofNet is drawn from popular undergraduate pure mathematics textbooks.
>
> Cross-benchmark transfer refers to evaluating how well a model trained on one benchmark performs when tested on a different benchmark without further fine-tuning. As evidenced in Table 3, state-of-the-art baselines like Kimina-Autoformalizer achieve near-saturation performance on MiniF2F (91\% LC+LSC@6) but experience a sharp decline on Gaokao-Formal (49\% LC+LSC@6). We observe a similar trend on PutnamBench, where the baseline achieves only 10\% LC+LSC@6. In contrast, our Mathesis-HPO model demonstrates superior transferability, reaching 30\% on PutnamBench (a 3x improvement). These significant performance gaps indicate that both Gaokao-Formal and PutnamBench introduce distinct distribution shifts and higher complexity that are not captured by existing benchmarks like MiniF2F.

---

> ### Author Response · Authors · 2025-11-21
> **Response to R4Q3**
>
> **Response to R4Q3**
>
> >_R4Q3: What safeguards ensure the reported Lean pass rates are not inflated by apply? /True/circular-goal artifacts?_
>
> Thanks for the question. To ensure that the reported Lean pass rates are not artificially inflated by artifacts such as apply?, we implement explicit post-verification safeguards. After a proof is generated, we re-parse and inspect the full proof script to detect the presence of these tactics. If such tactics are identified, the proof is discarded and counted as unsuccessful, regardless of whether Lean’s kernel accepts it.
>
> While we observe that a small number of proofs contain apply?, this is a phenomenon also noted in proofs generated by provers proposed in prior work (Ren et al., 2025). In our experiments, only proofs that both (1) successfully verify under Lean and (2) pass our artifact-filtering checks are counted as valid. We have added it to the Appendix. We highlight this issue to benefit the community and encourage more rigorous verification practices in future work.

---

### Official Review · Reviewer_6WKD · 2025-10-31

**Soundness:** 3
**Presentation:** 3
**Contribution:** 3
**Rating:** 6
**Confidence:** 3

**Summary:**

The paper introduces Mathesis, an end-to-end pipeline for formal theorem proving from natural language that centers on Mathesis-Autoformalizer (GRPO + DPO in a two-stage Hierarchical Preference Optimization), a semantic validation module LeanScorer (LLM-assisted subtask decomposition aggregated via a Sugeno fuzzy integral), and a new benchmark Gaokao-Formal (495 problems).

**Strengths:**

1. Formal theorem proving is a highly active research domain, and the motivation for the proposed system is clear and compelling.
2. The proposed methods are conceptually simple yet empirically effective.
3. The introduction of Gaokao-Formal is valuable in this field.

**Weaknesses:**

1. The paper lacks an adequate quality assessment of the Gaokao-Formal dataset. Some details are missing, such as annotator expertise, inter-annotator agreement statistics, and the protocol for resolving disagreements. Moreover, the subset used in Section 4.1 is not described, making it difficult to interpret the reported F1 scores.
2. LeanScorer introduces human priors and relies on additional compute (subtask decomposition → evaluation → aggregation) to produce higher scores. As such, it should be compared against other test-time scaling strategies, as well as alternative aggregation methods.

**Questions:**

1. Although LeanScorer improves semantic assessment, it may still produce erroneous judgments. What impact do these errors have on training stability or downstream performance?
2. The appendix notes that “the policy model is initialized from Kimina-Autoformalizer”, but this important detail should appear in the main text for clarity and reproducibility.

---

> ### Author Response · Authors · 2025-11-21
> **Response to R3W1 and R3W2**
>
> **Response to R3W1**
>
> >_R3W1: The paper lacks an adequate quality assessment of the Gaokao-Formal dataset. Some details are missing, such as annotator expertise, inter-annotator agreement statistics, and the protocol for resolving disagreements. Moreover, the subset used in Section 4.1 is not described, making it difficult to interpret the reported F1 scores._
>
> Thanks for the comment. We provide the following details regarding the quality assessment of the Gaokao-Formal dataset:
>
> Annotator Expertise:
> The annotation team consists of three highly qualified domain experts: an International Mathematical Olympiad (IMO) team member (Annotator 1) and two Lean formalization specialists (Annotator 2-3) from QS Top-10 mathematics departments. All annotators have extensive experience in both competitive mathematics and formal theorem proving.
>
> Inter-Annotator Agreement:
> We conducted a rigorous quality assessment on the subset used in Section 4.1, which contains 98 samples with independent triple annotation:
> - Fleiss' Kappa: 0.7545
> - Perfect three-way agreement: 81.63\% (80/98 samples)
> - Pairwise agreement rates:
>   - Annotator 1 vs 2: 92.86\%
>   - Annotator 1 vs 3: 83.67\%
>   - Annotator 2 vs 3: 86.73\%
>
> Disagreement Resolution Protocol:
> For the 18 samples (18.37\%) where annotators disagreed, we employed a consensus-based review process. The three annotators engaged in joint discussion sessions to reconcile differences, examining the formal Lean code and mathematical reasoning together until reaching unanimous agreement.
>
> The subset evaluated in Section 4.1 contains 98 problems, the original natural language questions are selected randomly from the complete dataset, and then parsed by an LLM formalizer to get a formal statement. The F1 scores reported reflect performance on this representative subset.
>
> We have added these details to Appendix G of our revised paper.
>
> **Response to R3W2**
>
> >_R3W2: LeanScorer introduces human priors and relies on additional compute (subtask decomposition → evaluation → aggregation) to produce higher scores. As such, it should be compared against other test-time scaling strategies, as well as alternative aggregation methods._
>
> Thanks for the comment. We would like to clarify that all the semantic-checking frameworks compared in Section 4.1 already incorporate a certain form of test-time scaling strategies. For LLM-as-a-Judge, following the literature, we applied a 4-time majority voting strategy and count a formalization as correct only if all four responses consistently give a positive judgment. The Re-informalization method involves two LLM calls for one round: one to translate the formal statement back into natural language, and the second compares it with the original question; and we also applied a 4-time majority voting with the same passing rule. So in total it cost 8 LLM calls. Our LeanScorer method requires two LLM calls: one for LLM-assisted subtask decomposition and another for subtask-level consistency annotation, followed by an additional aggregation step. This aggregation is computationally lightweight (within $6*10^{-3}$ seconds per 100 questions). Thus, the overall computational cost across the three methods is broadly comparable.
>
> We also thank the reviewer for the suggestion to compare alternative aggregation strategies. In the revised paper, we conducted a detailed comparison study between the Sugeno score and four simpler aggregation methods: binary, uniform averaging, geometric mean, and ordered weighted averaging. The results have been added to Appendix D. Our findings indicate that the Sugeno method achieves both the highest F1 score/accuracy and the most stable performance (lowest standard deviation of accuracy), supporting our choice of aggregation method.

---

> ### Author Response · Authors · 2025-11-21
> **Response to R3Q1 and R3Q2**
>
> **Response to R3Q1**
>
> >_R3Q1: Although LeanScorer improves semantic assessment, it may still produce erroneous judgments. What impact do these errors have on training stability or downstream performance?_
>
> Thanks for the question. LeanScorer, like all other LLM-based evaluation methods, may occasionally produce erroneous semantic judgments because LLM evaluators are subject to the inherent stochasticity, biases, and instability of large language models. We observe that these errors do not adversely affect training stability. To verify this, we monitored the reinforcement-learning process and found that the training loss evolves smoothly without signs of divergence or oscillation. We include the loss curve in the revised manuscript to illustrate this stability.
>
> In terms of the downstream performance, recall that our task begins with a natural-language mathematical statement, proceeds through its formalization, and ends with generating a Lean-verified proof that correctly resolves the original statement. In this setting, precision and recall have distinct implications. Specifically, high precision ensures that the statements admitted to the proving stage are semantically correct, thereby increasing the likelihood that the generated proofs correspond to the intended natural-language problems; however, this stricter filtering also reduces the number of statements passed to the prover and can therefore lower the overall proof success rate. High recall, in contrast, admits more candidate formalizations and can increase the prover’s apparent accuracy, but it also introduces a higher proportion of misaligned statements, which may yield Lean-verified proofs that do not resolve the original natural-language task.
>
> As shown in Section 4.1, among state-of-the-art LLM-based semantic checking methods, our LeanScorer achieves the best balance between precision and recall. Since the task requires producing a Lean-verified proof for the original natural-language problem, maintaining both high precision and high recall is essential for reliable overall performance. We have clarified this in Section 4.1 of the revised paper, marked in blue.
>
>
> **Response to R3Q2**
>
> >_R3Q2: The appendix notes that “the policy model is initialized from Kimina-Autoformalizer”, but this important detail should appear in the main text for clarity and reproducibility._
>
> Thank you for the suggestion. We have added this detail to the main text of the revised manuscript. Please refer to Section 3.1 (highlighted in blue).

---

> > ### Comment · Reviewer_6WKD · 2025-11-27
> > **Reply to Authors**
> >
> > Thanks for giving a clear point-by-point rebuttal to the weakness & questions I've raised. I think most of my questions are answered. As a result, I'm happy to increase my confidence score and the soundness assessment of this paper.

---

> > > ### Author Response · Authors · 2025-11-28
> > >
> > > We gratefully thank the reviewer for the positive feedback and for acknowledging the soundness of our research. We are glad that our responses addressed the questions raised and helped increase confidence in the positive assessment of our work.

---

### Official Review · Reviewer_brUb · 2025-10-31

**Soundness:** 4
**Presentation:** 4
**Contribution:** 4
**Rating:** 8
**Confidence:** 4

**Summary:**

This paper addresses the important and challenging problem of formal theorem proving directly from natural language. The authors point out that current state-of-the-art automated theorem provers (ATPs) mostly require formal input pre-written by human experts, which limits their real-world applicability. To solve this, the paper proposes Mathesis, a complete pipeline driven by "autoformalization." The core contributions of this work include: The first autoformalization model trained with Reinforcement Learning (RL), Mathesis-Autoformalizer, which innovatively combines syntactic, semantic, and prover feedback as reward signals; A novel semantic correctness evaluation framework, LeanScorer, for fine-grained assessment of formalized statements; A new, highly difficult benchmark dataset named Gaokao-Formal, containing 495 complex proof problems from China's National College Entrance Examination. Experimental results show that this method significantly outperforms existing techniques on multiple benchmarks and confirms the critical role of high-quality autoformalization in boosting the performance of downstream provers.

**Strengths:**

The combination of Reinforcement Learning with Hierarchical Preference Optimization (HPO) for the autoformalization task is the highlight of this paper. This approach enables dynamic learning from the feedback of a syntax checker, a semantic evaluator, and a downstream prover, significantly improving the quality of the formalized statements. The experimental results (e.g., a 45% relative improvement on Gaokao-Formal) fully demonstrate the effectiveness of this method.

The authors not only tested their method on two datasets with different difficulty levels and styles but also conducted extensive comparisons with a wide range of top-tier API models (including GPT-4o and Claude-3.5) and open-source models. More importantly, by pairing different autoformalizers with multiple mainstream provers, the paper clearly reveals the decisive impact of autoformalization quality on the final proof success rate.

The paper offers not just a model but a complete solution, including an innovative evaluation framework (LeanScorer) and a highly challenging new benchmark (Gaokao-Formal). These open-source resources will greatly facilitate subsequent development in the field.

**Weaknesses:**

The first is about the dependence of  LeanScorer on an LLM Evaluator. The core steps of LeanScorer (sub-task decomposition and consistency annotation) rely on an LLM. While experiments show high agreement with human annotators, this still introduces a potential source of noise, which to some extent undermines the credibility of the evaluation.

The second is that the paper uses downstream proof success as a metric may introduce bias. This could potentially lead the model to favor generating weakened versions of the statement and may weaken the credibility of the evaluation tests in section 4.3.

**Questions:**

1.	In the composite reward function for the GRPO stage, you perform a simple addition of the semantic reward $RsemR_{sem}Rsem​$ and the syntactic reward $RverR_{ver}Rver​$. This implies an assumption that they are equally important. Did you experiment with other aggregation methods, such as assigning different weights to them?
2.	Could you discuss whether the model has a potential tendency to generate weakened proofs?

---

> ### Author Response · Authors · 2025-11-21
> **Response to R2W1**
>
> **Response to R2W1**
>
> >_R2W1: The first is about the dependence of LeanScorer on an LLM Evaluator. The core steps of LeanScorer (sub-task decomposition and consistency annotation) rely on an LLM. While experiments show high agreement with human annotators, this still introduces a potential source of noise, which to some extent undermines the credibility of the evaluation._
>
> Thanks for the comment. At present, there are two viable families of semantic evaluators for autoformalization: (i) methods that rely on human-expert annotation or access to ground-truth formal statements, which are accurate but costly and non-scalable, and (ii) LLM-based automatic evaluation, which is scalable but inherits the stochasticity and biases of the underlying model, potentially introducing noise. LeanScorer, like the widely used LLM-as-a-Judge and Re-informalization methods, is LLM-based.
>
> The task we focus on requires evaluating autoformalization quality for real natural-language mathematical problems with available no ground-truth formal statements and thus requires high scalability. This makes human-expert annotation or ground-truth–based evaluators infeasible, leaving LLM-based methods as the practical option. However, current LLM-as-a-Judge and Re-informalization methods directly ask an LLM to provide a single binary holistic judgment in one prompt, which makes the evaluation highly sensitive to prompt phrasing and model stochasticity, provides no granularity for identifying which parts of the statement are misaligned, and often yields unstable or unreliable assessments for mathematically complex inputs.
>
> This motivates our LeanScorer to mitigate the above limitations. LeanScorer uses a multi-stage procedure consisting of LLM-assisted subtask decomposition, localized consistency annotation, and principled aggregation, reducing dependence on a single prompt and improving robustness to LLM variability. Empirically, LeanScorer exhibits high agreement with human annotations (Table 2), indicating that its design provides stable and credible semantic evaluations in practice.

---

> ### Author Response · Authors · 2025-11-21
> **Response to R2W2 and R2Q2**
>
> **Response to R2W2 and R2Q2**
>
> >_R2W2: The second is that the paper uses downstream proof success as a metric may introduce bias. This could potentially lead the model to favor generating weakened versions of the statement and may weaken the credibility of the evaluation tests in section 4.3._
>
> >_R2Q2: Could you discuss whether the model has a potential tendency to generate weakened proofs?_
>
> Thanks for the comment.
>
> As shown in Figure 2, the Lean compilation (syntactic check) and LeanScorer validation (semantic check) process safeguards against misaligned statements that could otherwise introduce bias into downstream proving: all formalized statements must pass both syntactic and semantic checks before entering the proving stage. Table 3 reports the quality of formalized statements produced by different models, including both Lean Check and Lean Check combined with LeanScorer Semantic Check. Figure 4 then evaluates theorem proving from natural language by pairing each autoformalizer with multiple downstream provers and measuring proof success rates. Importantly, proof accuracy in Figure 4 is computed only on statements that have already passed both syntax and semantic validation.
>
> To directly address the concern that our model might generate easier statements after DPO training, we conducted additional experiments with 100 randomly sampled problems (50 from Gaokao-Formal, 50 from MiniF2F) that passed LeanScorer validation.
>
>
> **Human Expert Evaluation** We ran a blind evaluation with a Lean 4 expert who assessed the semantic correctness of formalizations before and after DPO. Results are shown in the table below:
>
> | Dataset | Before DPO | After DPO |
> |---------|------------|-----------|
> | Gaokao-Formal | 70\% (35/50) correct | 78\% (39/50) correct |
> | MiniF2F | 90\% (45/50) correct | 92\% (46/50) correct |
>
> We observe that DPO training improves semantic correctness rather than sacrificing it for easier statements. This is consistent with our LC+LSC improvements in Table 3 (Mathesis 67\% to Mathesis-HPO 71\% on Gaokao-Formal, and from 25\% to 30\% on Putnam).
>
> **Prover-Based Difficulty Analysis** We recognize that objectively defining ``difficulty" for semantically correct statements is fundamentally challenging—without solving the problem, it is difficult to assess difficulty from the statement alone. As a proxy, we measured the average proof length (number of tactics) generated by DeepSeek-Prover-V2 7B (non-CoT mode, 128 samples per problem, excluding problems with no successful proofs). The metric is computed as:
>
> $$\text{Avg Proof Length} = \frac{1}{N} \sum_{i=1}^{N} \left( \frac{1}{|P_i|} \sum_{p \in P_i} \text{length}(p) \right)$$
>
> where $N$ is the number of problems and $P_i$ is the set of successful proofs for problem $i$. The results are as follows.
>
> | Dataset | Before DPO | After DPO |
> |---------|------------|-----------|
> | Gaokao-Formal | 31.26 | 33.40 |
> | MiniF2F | 24.97 | 25.44 |
>
> According to the results, proof complexity does not decrease after DPO. This does not support the hypothesis that DPO biases toward easier statements.
>
> **Case Study** We further provide one illustrative example (MiniF2F problem mathd_numbertheory_222), comparing the formalizations before and after DPO:
>
> Before DPO:
> ```
> theorem number_theory_969 :
>   {x : ℕ | x ≠ 120 ∧ Nat.lcm x 120 = 3720 ∧ Nat.gcd x 120 = 8} = {248}
> ```
>
> After DPO:
> ```
> theorem number_theory_447284 (x y : ℕ)
>   (h₀ : Nat.lcm x y = 3720) (h₁ : Nat.gcd x y = 8) (h₂ : x = 120) :
>   y = 248
> ```
>
> The above two statements are semantically equivalent to the natural-language statement, and the second formalization is easier to prove (integer equality vs. set equality). This reflects improved formalization, where the model produces a clearer and more canonical formulation that better matches Lean’s proof structure, which is precisely the goal of autoformalization, not a flaw.
>
> Overall, the results from our human expert evaluation, prover-based difficulty analysis, and qualitative examples support the conclusion that DPO improves semantic alignment rather than causing a tendency to generate weakened statements. We will release all evaluation data, including expert annotations and proof logs, for transparency and community verification.

---

> ### Author Response · Authors · 2025-11-21
> **Response to R2Q1**
>
> **Response to R2Q1**
>
> >_R2Q1: In the composite reward function for the GRPO stage, you perform a simple addition of the semantic reward and the syntactic reward  . This implies an assumption that they are equally important. Did you experiment with other aggregation methods, such as assigning different weights to them?_
>
> Thank you for the question. We adopted the unweighted summation strategy ($R = R_{sem} + R_{ver}$) based on the specific requirements of the autoformalization task and the nature of the reward signals, rather than theoretical invariance to differential weighting.
>
> **Joint Necessity and Binary Rewards**
> In our formulation (Section 3.1), both $R_{sem}$ (Semantic Correctness) and $R_{ver}$ (Syntactic Verification) are binary (0 or 1). Both are strictly necessary for successful autoformalization: a syntactically invalid statement cannot be processed by the prover, while a semantically incorrect statement solves the wrong problem.
>
> The unweighted sum naturally establishes a clear and desirable preference hierarchy:
> 1. R=2: Valid AND Correct (Ideal)
> 2. R=1: Valid XOR Correct (Partial success)
> 3. R=0: Invalid AND Incorrect (Failure)
> Assigning equal weights reflects this joint necessity and provides a strong, intuitive signal for the model to prioritize achieving both objectives simultaneously.
>
> **Empirical Effectiveness and Computational Efficiency**
> Reinforcement learning for reasoning tasks is computationally expensive. Introducing weights ($w_{sem}, w_{ver}$) would significantly expand the hyperparameter search space. Given the high cost of extensive tuning, we prioritized this robust, parameter-free approach.
>
> Empirically, this simple strategy proved highly effective. Mathesis-Autoformalizer-HPO achieved a 71\% pass rate on Gaokao-Formal, outperforming the previous state-of-the-art (49\%) by 22 percentage points (a 45\% relative gain). This confirms that the unweighted sum provided a sufficiently strong and stable signal to drive State-of-the-Art performance, although we agree that exploring optimized weighting schemes is a valuable direction for future work.

---

### Official Review · Reviewer_UUGT · 2025-11-01

**Soundness:** 2
**Presentation:** 2
**Contribution:** 2
**Rating:** 2
**Confidence:** 4

**Summary:**

The core contribution of this paper is (1) LeanScorer, a evaluation metric for semantic correctness in autoformalization and (2) training an autoformalization model using RL. The authors combined the autoformalization model with a formal theorem prover to prove statements in natural language.

**Strengths:**

* Evaluating the semantic correctness of autoformalized statements is an important bottleneck in autoformalization. The paper proposed LeanScorer that aims to address this problem
* This paper is one of the first to train autoformalization models with RL, which is nontrivial because of the difficulty in semantic evaluation.

**Weaknesses:**

* The task of "formal theorem proving from natural language" in this paper is not convincing. In this paper, it is defined as: Given a theorem statement in natural language, the model generates its formal statement and the corresponding formal proof. It is evaluated using proof accuracy (Fig. 4). This task definition does not evaluate whether the formalized statement is semantically correct. The model could formalize the natural language statement into a easier formal statement so that the proof accuracy will be higher. I understand that the authors have evaluated their autoformalizer, and experiments show improvements compared to existing autoformalizers under the LeanScorer metric defined in this paper (Table 3). However, that does not resolve this issue. It could be that the autoformalizer in this paper is indeed more accurate but is biased towards making statements easier, whereas previous autoformalzers are less accurate but are biased towards making statements harder. If true, this could explain the results in Fig. 4.

*  The comparison between LeanScorer and LLM-as-a-Judge (Table 2) is inconclusive. LeanScorer has better precision whereas LLM-as-a-Judge has better recall. The F1 score is not a good metric because it assumes precision and recall are equally important. When evaluating autoformalized statements, however, precision is arguably less important than recall, as incorrectly translated statements can also be useful for training theorem provers. It would be great to compare LeanScorer and LLM-as-a-Judge in terms of their downstream impact to theorem proving.

* Need ablation studies to justify the design of LeanScorer, e.g., why Sugeno Fuzzy Integral instead of simpler methods for computing the overall score.

* MiniF2F is already saturated (SOTA methods are close to 100%). It would be great to add results on PutnamBench.

**Questions:**

N/A

---

> ### Author Response · Authors · 2025-11-21
> **Response to R1W1**
>
> **Response to R1W1**
>
> >_R1W1: The task of "formal theorem proving from natural language" in this paper is not convincing. ... This task definition does not evaluate whether the formalized statement is semantically correct. The model could formalize the natural language statement into a easier formal statement so that the proof accuracy will be higher. I understand that the authors have evaluated their autoformalizer, and experiments show improvements compared to existing autoformalizers under the LeanScorer metric defined in this paper (Table 3). However, that does not resolve this issue. It could be that the autoformalizer in this paper is indeed more accurate but is biased towards making statements easier, whereas previous autoformalzers are less accurate but are biased towards making statements harder. If true, this could explain the results in Fig. 4._
>
> Thanks for the comment.
>
> We would like to clarify that our task definition does evaluate semantic correctness. As shown in Figure 2, all formalized statements must pass both Lean compilation (syntactic check) and LeanScorer validation (semantic check) before entering the proving stage. This safeguards against misaligned statements that could otherwise introduce bias into downstream proving. Table 3 reports the quality of formalized statements produced by different models, including both Lean Check and Lean Check combined with LeanScorer Semantic Check. Figure 4 then evaluates theorem proving from natural language by pairing each autoformalizer with multiple downstream provers and measuring proof success rates. Importantly, proof accuracy in Figure 4 is computed only on statements that have already passed both syntax and semantic validation. In other words, the proof accuracy in Figure 4 directly reflects the quality of the autoformalizations.
>
> To directly address the concern that our model might generate easier statements after DPO training, we conducted additional experiments with 100 randomly sampled problems (50 from Gaokao-Formal, 50 from MiniF2F) that passed LeanScorer validation.
>
> **Human Expert Evaluation** We ran a blind evaluation with a Lean 4 expert who assessed the semantic correctness of formalizations before and after DPO. Results are shown in the table below:
>
> | Dataset | Before DPO | After DPO |
> |---------|------------|-----------|
> | Gaokao-Formal | 70\% (35/50) correct | 78\% (39/50) correct |
> | MiniF2F | 90\% (45/50) correct | 92\% (46/50) correct |
>
> We observe that DPO training improves semantic correctness rather than sacrificing it for easier statements. This is consistent with our LC+LSC improvements in Table 3 (Mathesis 67\% to Mathesis-HPO 71\% on Gaokao-Formal, and from 25\% to 30\% on Putnam).
>
> **Prover-Based Difficulty Analysis** We recognize that objectively defining ``difficulty" for semantically correct statements is fundamentally challenging—without solving the problem, it is difficult to assess difficulty from the statement alone. As a proxy, we measured the average proof length (number of tactics) generated by DeepSeek-Prover-V2 7B (non-CoT mode, 128 samples per problem, excluding problems with no successful proofs). The metric is computed as:
>
> $$\text{Avg Proof Length} = \frac{1}{N} \sum_{i=1}^{N} \left( \frac{1}{|P_i|} \sum_{p \in P_i} \text{length}(p) \right)$$
>
> where $N$ is the number of problems and $P_i$ is the set of successful proofs for problem $i$. The results are as follows.
>
> | Dataset | Before DPO | After DPO |
> |---------|------------|-----------|
> | Gaokao-Formal | 31.26 | 33.40 |
> | MiniF2F | 24.97 | 25.44 |
>
> According to the results, proof complexity does not decrease after DPO. This does not support the hypothesis that DPO biases toward easier statements.
>
> **Case Study** We further provide one illustrative example (MiniF2F problem \texttt{mathd\_numbertheory\_222}), comparing the formalizations before and after DPO:
>
> Before DPO:
>
> ```
> theorem number_theory_969 :
>   {x : ℕ | x ≠ 120 ∧ Nat.lcm x 120 = 3720 ∧ Nat.gcd x 120 = 8} = {248}
> ```
>
>
> After DPO:
> ```
> theorem number_theory_447284 (x y : ℕ)
>   (h₀ : Nat.lcm x y = 3720) (h₁ : Nat.gcd x y = 8) (h₂ : x = 120) :
>   y = 248
> ```
>
> The above two statements are semantically equivalent to the natural-language statement, and the second formalization is easier to prove (integer equality vs. set equality). This reflects improved formalization, where the model produces a clearer and more canonical formulation that better matches Lean’s proof structure, which is precisely the goal of autoformalization, not a flaw.
>
> Overall, the results from our human expert evaluation, prover-based difficulty analysis, and qualitative examples support the conclusion that DPO improves semantic alignment rather than causing a tendency to generate weakened statements. We will release all evaluation data, including expert annotations and proof logs, for transparency and community verification.

---

> ### Author Response · Authors · 2025-11-21
> **Response to R1W2, R1W3, and R1W4**
>
> **Response to R1W2**
>
> >_R1W2: The comparison between LeanScorer and LLM-as-a-Judge (Table 2) is inconclusive. LeanScorer has better precision whereas LLM-as-a-Judge has better recall. The F1 score is not a good metric because it assumes precision and recall are equally important. When evaluating autoformalized statements, however, precision is arguably less important than recall, as incorrectly translated statements can also be useful for training theorem provers. It would be great to compare LeanScorer and LLM-as-a-Judge in terms of their downstream impact to theorem proving._
>
> Thanks for the comment.
>
> Semantic check verifies whether the generated formal statement preserves the intended meaning of the original natural language input. LeanScorer and LLM-as-a-Judge are two alternative methods for performing this semantic check. As commonly adopted by the literature, we use both recall and precision to evaluate how these methods align with human experts. High precision means that statements a method accepts as correct are very likely to be truly correct. High recall means that the method successfully identifies most of the statements that are truly semantically correct.
>
> However, recall and precision each introduce their own bias: recall rewards high coverage even at the cost of false positives, whereas precision rewards strict selectivity even at the cost of false negatives. Thus, we report the F1 score, which provides a balanced harmonic mean of them.
>
> LLM-as-a-Judge has 100\% recall and 73\% precision, indicating that it is a less selective semantic checker, which may pass many false-positive statements whose meaning differs from the original problem. Under our task setting—“the task begins with a given natural language statement and ends with the generation of a formal proof”—such false positives lead to irrelevant proofs, because proving an irrelevant statement provides no value regardless of whether the proof succeeds. For this reason, LLM-as-a-Judge is insufficient as a standalone semantic checker for this autoformalization–proving pipeline. Motivated by this observation, we propose LeanScorer that enables a more fine-grained and task-adaptive assessment.
>
> We agree that a high-recall checker will pass more statements and thus yield more training data. However, the task addressed in this paper requires a stricter semantic checker to filter out misaligned formalizations, ensuring that the generated proofs correspond to and correctly solve the intended input problems. We've clarified this explicitly in the revised paper.
>
> **Response to R1W3**
>
> >_R1W3: Need ablation studies to justify the design of LeanScorer, e.g., why Sugeno Fuzzy Integral instead of simpler methods for computing the overall score._
>
> Thanks for the suggestion. We conducted a detailed comparative study between the Sugeno score and four simpler aggregation methods: binary, uniform averaging, geometric mean, and ordered weighted averaging. The experimental results indicate that the Sugeno method achieves both the highest F1 score/accuracy and the most robust performance. The results and analysis have been added to Appendix D of our revised paper.
>
> **Response to R1W4**
>
> >_R1W4: MiniF2F is already saturated (SOTA methods are close to 100\%). It would be great to add results on PutnamBench._
>
> Thanks for the suggestion. We have conducted experiments on PutnamBench and incorporated the results into the revised manuscript. The new evaluations are included in Table 3 and highlighted in \blue{blue}. Entries marked TBD will be updated once the experiments are complete, and we will further integrate PutnamBench into our analysis. As shown in the results, both Mathesis variants consistently outperform Herald-Autoformalizer and Kimina-Autoformalizer on PutnamBench across LC and LC+LSC, consistant with the results on MiniF2F and Gaokao-Formal. In particular, Mathesis-Autoformalizer-HPO achieves the strongest results, reaching 73\% LC and 30\% LC+LSC at k=6, and 38\% LC and 10\% LC+LSC at k=1. This demonstrates the effectiveness of our approach.

---

> > ### Comment · Reviewer_UUGT · 2025-11-25
> >
> > Thank you for the detailed response. It helps address many of my concerns, and I'd like to raise the score to 6.

---

> > > ### Author Response · Authors · 2025-11-28
> > >
> > > We sincerely thank the reviewer for the positive feedback and for raising the score. We are pleased that our clarifications and additional experiments have helped address the reviewer’s concerns and further strengthened our work.

---

### Author Response · Authors · 2025-12-03
**Summary of Revision**

Dear Area Chair and Reviewers,

We sincerely thank the reviewers for their time and constructive comments, which have significantly improved our work. For the AC, we provide a summary of our rebuttal below.

**Paper Overview.**
We introduce Mathesis, the first pipeline for the systematic study of formal theorem proving directly from natural language. We contributed (1) Mathesis-Autoformalizer, the first autoformalizer trained with reinforcement learning integrating syntactic, semantic, and prover feedback; (2) LeanScorer, a novel framework for fine-grained semantic evaluation; and (3) Gaokao-Formal, a new benchmark of challenging complex proof problems.

**Score Trajectory.**
We respectfully wish to emphasize the significant positive trajectory of our paper during the discussion. We refer Reviewer UUGT, brUb, 6WKD, UPch, and NXS9 as R1, R2, R3, R4, and R5, respectively. Three reviewers responded timely and all provided positive responses and raised the scores:
  * R1 (UUGT): **ranking score 2→6** -- "Thank you for the detailed response. It helps address many of my concerns, and **I'd like to raise the score to 6**."
  * R5 (NXS9): **ranking score 4→6** -- "I think the authors have adequately responded to most of my concerns… **I am satisfied** with the expanded explanations… I have revised my score."
  * R3 (6WKD): **confidence 3→4, soundness 3→4** -- "Thanks for giving a clear point-by-point rebuttal… **I'm happy to increase** my confidence score and the soundness assessment…."

This brings our scores to **[8, 6, 6, 6, 6]**, reflecting the successful resolution of the initial concerns and the recognition of the novelty and significance of our work.

**Summary of Concerns and Responses.**
We have carefully addressed all concerns raised by the reviewers and incorporated additional experiments, analyses, clarifications, and revisions into the updated manuscript.

1. Potential Bias in Formalizations: Reviewers raised concerns that training might bias the autoformalizer toward generating semantically incorrect but easier-to-prove weakened statements. In response, we (1) clarified how our pipeline safeguards against errors through rigorous syntactic and semantic validation, (2) provide human expert quality assessment, difficulty analysis, and case studies (Appendix H). Results show that we successfully trained the model to generate prover-friendly statements while preserving original semantic content and problem difficulty. [R1W1, R2W2, R2Q2, R5W4].

2. LeanScorer Design and Validation: Reviewers questioned LeanScorer's design, advantages over baselines, and validation details. We (1) clarified its architecture and how it differs from and improves upon prior baselines; (2) explained its LLM dependence and how it mitigates potential noise, with experimental support; and (3) detailed the annotation protocol and quality-control procedures (Appendix G) in constructing the validation set. [R1W2, R2W1, R3W1, R3Q1, R4W2, R5W2, R5Q3, R5Q5].

3. Data Contamination and Benchmarking: Reviewers raised concerns about potential overlap between training and benchmark data, and details of the Gaokao-Formal benchmark. We (1) performed N-gram overlap analysis (Appendix B), confirming zero "dirty" matches between training and testing data; (2) clarified that the training and Gaokao-Formal data are distinct sources; (3) detailed Gaokao-Formal's difficulty and topic mix relative to existing benchmarks (Appendix C). [R3W1, R4W1, R4Q2, R5W3, R5Q1].

4. Ablation Study on LeanScorer Aggregation Method: To validate LeanScorer's aggregation method, we added (1) ablation studies comparing Sugeno integral aggregation to alternative methods (Appendix D.1), and (2) comparison of test-time scaling strategies. Results confirm our chosen aggregation method provides very strong alignment with human semantic judgments.  [R1W3, R3W2, R5W1].

5. Additional Experiments and Presentation: Reviewers suggested adding a more challenging benchmark and additional baselines, and improving the presentation. We (1) added results on PutnamBench (Table 3), showing Mathesis-HPO largely outperforms baselines with strong cross-domain generalization; (2) evaluated additional baselines confirming our superior performance; and (3) refined the presentation. [R1W4, R4Q1, R2Q1, R3Q2, R4Q3, R5W5, R5Q2, R5Q4, R5Q6].

Best regards,

The Authors

---

### Meta-Review · Area_Chair_9TTS · 2025-12-05

**Summary:**

This paper gives a treatment to auto-formalisation, and it seems to improve it significantly. I believe the paper is well-written and addresses an important problem, and the reviewers seem to point to several of its strengths. Given the incident with OpenReview, it is hard to judge the score increases, but I believe even apriori the paper is relatively well-received.

**Reviewer Concerns:**

The last reviewer concern might still stand, in regards to a simple baseline that could obtain comparable results.

**Reviewer Scores:**

It is hard to judge, I cannot predict such a thing.

---

### Decision · Program_Chairs · 2026-01-26

Accept (Poster)